# MoE-CAP: Benchmarking Cost, Accuracy and Performance of Sparse Mixture-of-Experts Systems

**Yinsicheng Jiang**[1*]   **Yao Fu**[1*]   **Yeqi Huang**[1*]   **Ping Nie**[3]   **Zhan Lu**[1]   **Leyang Xue**[1]

**Congjie He**[1]   **Man-Kit Sit**[1]   **Jilong Xue**[2]   **Li Dong**[2]   **Ziming Miao**[2]

**Dayou Du**[1]   **Tairan Xu**[1]   **Kai Zou**[4]   **Edoardo Ponti**[1,5]   **Luo Mai**[1]

[1]University of Edinburgh   [2]Microsoft Research   [3]Peking University   [4]NetMind.AI   [5]NVIDIA

## Abstract

The sparse Mixture-of-Experts (MoE) architecture is increasingly favored for scaling Large Language Models (LLMs) efficiently, but it depends on heterogeneous compute and memory resources. These factors jointly affect system Cost, Accuracy, and Performance (CAP), making trade-offs inevitable. Existing benchmarks often fail to capture these trade-offs accurately, complicating practical deployment decisions. To address this, we introduce MoE-CAP, a benchmark specifically designed for MoE systems. Our analysis reveals that achieving an optimal balance across CAP is difficult with current hardware; MoE systems typically optimize two of the three dimensions at the expense of the third—a dynamic we term the MoE-CAP trade-off. To visualize this, we propose the CAP Radar Diagram. We further introduce sparsity-aware performance metrics—Sparse Memory Bandwidth Utilization (S-MBU) and Sparse Model FLOPS Utilization (S-MFU)—to enable accurate performance benchmarking of MoE systems across diverse hardware platforms and deployment scenarios. Our code has been released at: `https://github.com/Auto-CAP/MoE-CAP`

## 1   Introduction

Recent large language models (LLMs) are increasingly adopting sparse Mixture-of-Experts (MoE) architectures, notable examples of which include Switch-C [14], DBRX, Mixtral-8x22B [25], Snowflake Arctic [39], Grok-1 [46], DeepSeek-MoE [10], and Qwen1.5-MoE [4]. These models utilize sparse experts grouped into an MoE layer, and these experts are selectively activated through a router (or a gating network). By routing tokens to a subset of experts, MoEs achieve sub-linear computational costs compared to their dense equivalents, which allows building trillion-parameter-scale LLMs.

Current MoE systems exhibit increasing complexity, driven by two main factors: (i) There is enhanced sophistication in the design of sparse MoE layers and gating networks (or routers), which differ in sparsity characteristics across various MoE models—**we define sparsity in MoE systems as the ratio of activated to total parameters per token**; (ii) MoEs demonstrate sub-linear computational complexity, enabling the offloading of less frequently activated experts onto external memory and processors. This approach reduces dependence on costly High Bandwidth Memory (HBM) on GPUs. Consequently, the complexity of servers hosting MoE systems has escalated, with these servers typically featuring heterogeneous compute, memory, and communication resources, arranged in a

---

[*]Co-leading authors.

39th Conference on Neural Information Processing Systems (NeurIPS 2025) Track on Datasets and Benchmarks.

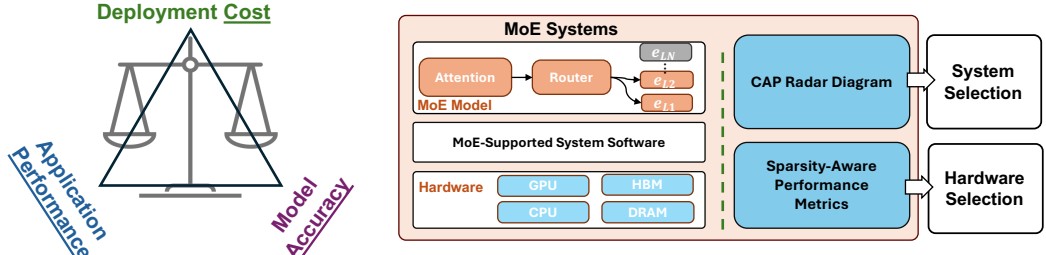

Figure 1: Overview of MoE-CAP. **Left**: We identify trade-offs between hardware Cost, model Accuracy, and application Performance. **Right**: MoE-CAP introduces new sparsity-aware metrics and CAP radar diagrams to accurately and comprehensively evaluate MoE systems, helping users choose both the right MoE system and suitable hardware.

multi-tier architecture. For instance, modern MoE systems increasingly offload experts to DRAM and SSDs [49, 13] and delegate part of the computation to CPUs [26].

Practitioners of MoE systems are actively seeking methods to benchmark their cost, accuracy (in downstream tasks), and performance (time and memory efficiency) in order to optimize their deployment. However, this benchmarking is challenging for the following reasons: (i) *Poor understanding of the relation between cost, accuracy and performance*: real-world deployments of MoE systems frequently reveal underestimated costs, under-achievements in performance benefits, and compromised accuracy. Clear principles are needed to help practitioners effectively evaluate and understand the complex interplay between these factors, (ii) *Inadequate system cost and performance assessment metrics*: Existing metrics like Memory Bandwidth Utilization (MBU) [1] and Model FLOPS Utilization (MFU) [8] fail to account for the sparse activation patterns of experts in MoE systems. This oversight leads to overestimated memory and compute costs. Additionally, current benchmarks predominantly estimate costs based on GPU usage alone. However, modern MoE systems increasingly rely on heterogeneous processors and multi-tier memory resources. Ignoring these factors yields inaccurate cost estimations.

To address the above issues, this paper introduces MoE-CAP, a benchmark designed to evaluate and understand the cost, accuracy, and performance of MoE systems. The design of MoE-CAP (illustrated in Figure 1) offers several key contributions:

**(1) A benchmarking method for understanding MoE system trade-offs.** We analyze a broad range of MoE systems and observe that their optimizations typically compromise one of three key properties—cost, accuracy, or performance—while prioritizing the other two. Based on this observation, we categorize MoE systems into three types: cost–performance optimized, accuracy–cost optimized, and accuracy–performance optimized. To better capture and compare these trade-offs, we introduce the CAP radar diagram, a benchmarking method that highlights the strengths and limitations of each system, helping users select the most suitable MoE system based on their deployment needs.

**(2) Sparsity-aware performance metrics.** We propose two sparsity-aware performance metrics: Sparse Memory Bandwidth Utilization (Sparse MBU) and Sparse Model FLOPS Utilization (Sparse MFU). These metrics enable accurate predictions of the compute and memory bandwidth savings achievable with MoE systems. As a result, they serve as a valuable guide for determining whether lower-power, cost-efficient processors can effectively support large MoE models without performance bottlenecks. This provides a formal explanation for how recent models such as DeepSeek-R1 significantly reduce the reliance on expensive, high-performance processors.

**(3) Comprehensive benchmark implementation and coverage.** We developed an automated workflow to evaluate MoE systems on current and emerging hardware using sparsity-aware metrics and comparing them via the CAP radar diagram. It supports diverse MoE models—including QWen3 and DeepSeek-R1—and enables evaluation across multiple datasets and MoE-serving systems such as SGLang, vLLM, K-Transformer and MoE-Infinity.

Table 1: Characteristics of recent open-source MoE models.

| Model | Total Param | Active Param | # of Experts | Top-k + Shared |
|---|---|---|---|---|
| Switch-C | 1571B | 12B | 128 | 1 |
| DBRX | 132B | 36B | 16 | 4 |
| Mistral-8x22B | 141B | 39B | 8 | 2 |
| Snowflake Arctic | 480B | 17B | 128 | 2 |
| Grok-1 | 314B | 77B | 8 | 2 |
| DeepSeek-R1 | 671B | 37B | 256 | 8 + 1 |
| Qwen1.5-MoE | 14.3B | 2.7B | 60 | 4 + 4 |
| Moonlight-16B-A3B | 16B | 3B | 64 | 6 + 2 |
| Qwen3-235B-A22B | 235B | 22B | 128 | 8 |

## 2 Background and Motivation

**Sparse MoE models.** Many sparse MoE models have been designed recently, and their characteristics are summarized in Table 1. These models typically possess a large number of parameters, with some reaching as high as 1571 billion. During token processing, only a subset of these parameters, generally between 1–25%, is activated. MoE models display diverse sparsity characteristics. For instance, Mistral-8x22B features large experts, each containing 0.3 billion parameters, but restricts the number of experts per layer to just 8. In contrast, Snowflake Arctic uses a much higher number of experts per layer (128), though each expert is smaller, containing only 0.15 billion parameters. The complexity extends to the router design, which typically selects the top-K experts (where K usually ranges from 1 to 4) for activation. Some models also employ a hybrid approach, incorporating a set of always-activated "shared" experts along with a selectively activated group of K experts. These shared experts may differ in size from the selectively activated ones, adding another layer of complexity to the model architecture.

**Emerging MoE system designs.** MoE systems exhibit sub-linear computational complexity but still require increased memory to accommodate all potentially activated experts. To enhance memory efficiency, several system designs have been explored recently: (i) Designs applying quantization to experts, such as unified quantization for all experts—e.g., GPTQ [16], AWQ [32], and SmoothQuant [47]—and expert-specific adaptive quantization—e.g., QMoE [15] and MoQE [27]. (ii) Designs offloading experts to external memory. Examples include MoE-layer-wise parameter offloading—e.g., DeepSpeed-Inference [2] and SwapAdvisor [23]—and fine-grained expert-level offloading—e.g., MoE-Infinity [49], Brainstorm [9], Mixtral-Offloading [13]. (iii) Designs utilizing CPU cores for sharing partial computation from GPUs, such as computing less input-intensive experts on CPUs— e.g., Fiddler [26].

**Heterogeneous resources in MoE systems.** The sparse nature of MoE models has led system designers to increasingly utilize heterogeneous resources for hosting sparse experts, thus enhancing the cost–performance ratio of these systems. These resources are typically more cost-effective than their GPU counterparts, while still offering significant computing power and memory bandwidth, sufficient for certain operations in MoE models. These resources include: (i) Heterogeneous compute resources, such as CPUs integrated within GPUs (e.g., Grace-Hopper ARM chips) and external CPUs (e.g., AMD and Intel x86 chips). (ii) Heterogeneous memory resources, including LPDDR and HBM in a Grace-Hopper Superchip, DRAM in host CPU, and high-speed SSDs. (iii) Heterogeneous communication resources, featuring chip-to-chip links for HBM-DRAM within GPUs, PCIe for external GPU communications, and NVLink for inter-GPU communication.

### 2.1 Why existing benchmarks fall short with MoE systems

Many LLM benchmarks have been proposed over the past few years, including MLPerf [38], ML-Energy [51], Open-LLM-Leaderboard [5], LLM-Perf [24], TensorDock [40], and Artificial Analysis [3]. To evaluate LLM systems, these benchmarks primarily use three types of metrics: (i) overall system performance metrics, such as token throughput, prefilling time, and decoding time, (ii) cost metrics, like tokens-per-dollar and tokens-per-kilowatt, focusing solely on GPU usage, and (iii) fine-grained system performance metrics, such as MBU and MFU, to facilitate the understanding

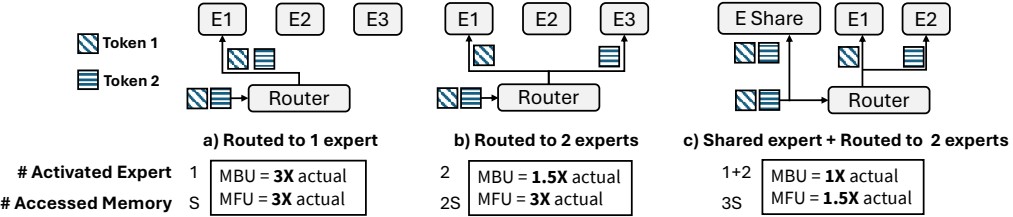

Figure 2: MoE memory access and performance metrics under three routing scenarios. $S$ denotes the size of a single expert. Existing MBU/MFU definitions overestimate costs by ignoring routing and expert selection. We show the extent of this overestimation relative to actual values.

of memory and compute bottlenecks, as discussed in recent MoE articles [1]. However, when these benchmarks are applied to MoE systems, they demonstrate several limitations.

**(1) Lack of principles to understand the relation between cost, accuracy and performance in MoE systems.** While MoE systems often claim advantages in cost, accuracy, and performance, their real-world deployment frequently reveals significant challenges. Practitioners often find that deployment costs are underestimated, promised performance benefits over dense alternatives are not achieved, and model accuracy is often compromised. There is a growing need for clear principles to help users effectively assess and understand the complex interplay between these factors.

**(2) Inaccurate system cost and performance assessment metrics.** The commonly used MBU [1] and MFU [8] metrics fail to consider the selective activation within sparse MoE layers. MBU measures memory bandwidth utilization as $\text{MBU} = \frac{B_{\text{achieved}}}{B_{\text{peak}}} = \frac{(S_{\text{model}} + S_{\text{KV}})/\text{TPOT}}{B_{\text{peak}}}$, where $S_{\text{model}}$ represents the full model size and $S_{\text{KV}}$ is the KV cache size, $B_{\text{achieved}}$ refers to the achieved memory bandwidth per forward pass, $B_{\text{peak}}$ is the peak bandwidth of the underlying hardware. Similarly, MFU measures compute utilization as $\text{MFU} = (T_{\text{token}} \times F_{\text{token}})/F_{\text{peak}}$ , assuming all parameters participate in computation, where $T_{\text{token}}$ is the token throughput, $F_{\text{token}}$ represents the FLOPs required per token, and $F_{\text{peak}}$ indicates the peak FLOPs capacity of the hardware. However, as illustrated in Figure 2, these metrics significantly overestimate resource utilization by assuming all experts are active, especially when batch size >1. For example, when only one expert out of three is activated (case a), both MBU and MFU are inflated by 3×. Even with shared experts (case c), the metrics still overestimate utilization by 1.5×. This often leads operators to substantially over-provision GPUs, causing significant resource waste.

## 3 The CAP Benchmarking Method

We propose the following **CAP benchmarking method** for understanding, benchmarking, and comparing MoE systems. The CAP method encompasses three key system optimization metrics:

- **Cost (C):** We propose a cost model for MoE systems that accounts for all hardware components, reflecting the increasing use of heterogeneous resources (e.g., CPUs and GPUs). Unlike existing benchmarks that consider GPU cost in isolation, our model jointly evaluates the cost—measured in power or monetary terms—of computation, communication, and memory. The computation cost includes both CPU and GPU usage. Communication cost captures system interconnects (e.g., PCIe links and NVLink). Memory cost spans multiple tiers (i.e., HBM, DRAM, PCIe SSD and emerging CXL).
- **Accuracy (A):** Accuracy in MoE systems is broadly defined to encompass a range of evaluation metrics commonly used in prominent LLM benchmarks, such as the Open LLM Leaderboard [5], which reflect real-world applications of large language models. To support comprehensive task evaluation—spanning multiple-choice, reasoning, and QA tasks—we adopt diverse metrics including exact match, F1 score, and win rate, ensuring broad coverage across various evaluation scenarios.
- **Performance (P):** Performance metrics in MoE systems vary by deployment scenario. In online serving, time-per-output-token (TPOT) is crucial, reflecting the need for low decoding latency. It is influenced by factors like memory bandwidth and FLOPS. In contrast, offline inference and training prioritize token throughput and end-to-end latency for faster task completion. We support

switching between metrics such as TPOT, throughput, bandwidth utilization, and FLOPS to suit different use cases.

## 3.1 Complete cost model of MoE systems

We introduce the design of our cost model, which aims to capture the complexity of managing various heterogeneous resources in an MoE system.

**Purchase cost.** Upon server decommissioning and during model deployment upgrades, new hardware purchases are factored into the overall cost. From a performance perspective, system choice is influenced by three main components as outlined in Equation 1: computation, communication, and memory capacity. The computation component includes both the GPU and CPU. Communication involves the connectivity within the system, such as the PCIe links between the GPU, CPU, and SSD, and the NVLink connections between GPUs. We also consider the communication between the computation units and memory, specifically CPU-to-DRAM and SRAM-to-HBM in GPUs. Memory is structured in three hierarchical tiers: HBM within the GPU, DRAM as host memory, and SSD storage. More formally, we define Equation 1 as follows where $C$ stands for the dollar cost for the target hardware:

$$C_{\text{hardware}} = \overbrace{(C_{\text{GPU}} + C_{\text{CPU}})}^{\text{computation}} + \overbrace{(C_{\text{C2M}} + C_{\text{PCIe}} + C_{\text{NVLink}})}^{\text{communication}} + \overbrace{(C_{\text{HBM}} + C_{\text{DRAM}} + C_{\text{SSD}})}^{\text{memory}}, \quad (1)$$
$$= C_{\text{GPU}} + C_{\text{CPU}} + C_{\text{Motherboard}} + C_{\text{DRAM}} + C_{\text{SSD}},$$

Specifially, $C_{\text{C2M}}$ is the accelerator internal bandwidth cost. Choosing the GPU variant $C_{\text{GPU}}$ involves the options for HBM size $C_{\text{HBM}}$, communication links $C_{\text{NVLink}}$ and $C_{\text{C2M}}$. Choosing the motherboard variant $C_{\text{Motherboard}}$ covers the PCIe cost. Choosing the CPU variant also caps the $C_{\text{C2M}}$.

**Energy cost.** After the server is purchased and the model is deployed, the primary cost comes from using the hardware, which is reflected as energy consumption. (shown in Equation 2). The energy consumption primarily stems from using the data on the device (computation) and moving the data (communication) between and within devices. The maintenance power draw on each layer of memory is included as the basis for computation and communication. We account for the average power draw over the server runtime $R$.

$$C_{\text{energy}} = \left[ \overbrace{(P_{\text{GPU}} + P_{\text{CPU}})}^{\text{computation}} + \overbrace{(P_{\text{C2M}} + P_{\text{PCIe}} + P_{\text{NVLink}})}^{\text{communication}} \right] \times R \quad (2)$$

**Cost-performance.** The performance metric (e.g., tokens-per-second throughput) only measures whether the model achieves a guaranteed performance on a given server setting. However, cost efficiency has yet to be considered: how to identify the best server setup to achieve the desired performance? Cost efficiency must take into account both the purchase cost and the energy cost of the model deployment.

$$C_{\text{token}} = (C_{\text{hardware}} + C_{\text{energy}} \times \$/\text{kWh})/(T_{\text{token}} \times R), \quad (3)$$

We apply Equation (3) to combine both cost models and performance metrics. We aim to determine the per-token cost in units of dollars, denoted as $C_{\text{token}}$. The energy cost in kWh needs to be combined with the local energy price to estimate the cost, as it can vary between industrial/personal energy usage and by region [41]. The $C_{\text{token}}$ is calculated over the lifetime $R$ of the deployment and averaged across all generated tokens. To further explain our cost model, we illustrate the use cases of our cost model in Appendix A.2.

## 3.2 Understanding strengths of MoE systems through CAP analysis

While most MoE systems claim strengths in performance, cost, and accuracy, they typically optimize only two at the expense of the third. Based on this, we group them as follows:

**MoE systems for Performance and Accuracy (PA).** Improving performance without compromising accuracy can be achieved through two primary approaches: (i) scaling up device memory capacity

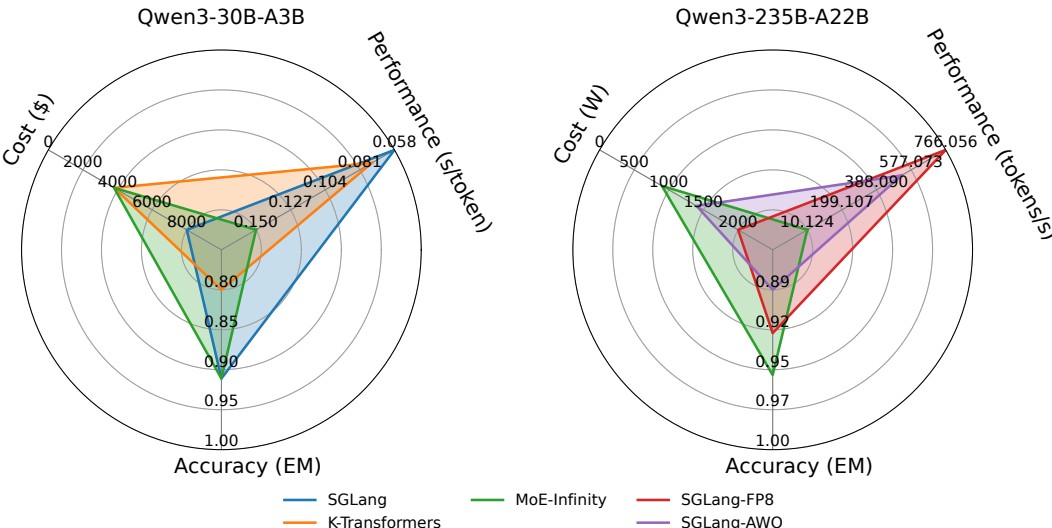

Figure 3: **CAP Radar diagrams** comparing representative PA, PC, and CA systems. **Left:** Trade-offs among SGLang (PA), K-Transformers (PC), and MoE-Infinity (CA) on Qwen3-30B-A3B. **Right:** Trade-offs comparing with offloading (MoE-Infinity) and quantization (SGLang-FP8/AWQ).

by utilizing high-end GPUs with large memory (e.g., H200 NVL with 141 GB and MI300X with 192 GB), and (ii) scaling out memory capacity via parallel and distributed computing, leveraging technologies such as NVLinks, NVSwitch, and InfiniBand, along with techniques like Tensor, Pipeline, and Expert parallelism [30, 21, 31, 28, 37]. However, these approaches face significant drawbacks: the exponential increase in system costs due to the growing manufacturing complexity of high-speed memory devices and the sub-linear scaling efficiency of distributed systems, where communication increasingly becomes a bottleneck at larger scales.

**MoE systems for Performance and Cost (PC).** To enhance system performance without increasing costs, effective model compression techniques such as quantization and sparsification can be adopted for MoE models. Specifically, quantization reduces model size by mapping parameters to lower precisions, such as 8-bit [11, 50], 4-bit [32, 16], or even 2-bit [12, 7], and accelerates latency through advanced system implementations [17, 44] featuring optimized data layouts and efficient dequantization. Although MoE models inherently exhibit sparsity, their application in long-context scenarios faces challenges due to the quadratic complexity of attention mechanisms. Sparse attention, therefore, emerges as a compelling solution for MoE models [48, 19, 33, 52]. However, compression typically introduces accuracy degradation due to the limited representational capacity of compressed parameters.

**MoE systems for Cost and Accuracy (CA).** In constrained hardware environments, some systems aim to maintain model accuracy within a limited budget (e.g., relying on low-end GPUs or a restricted number of GPUs). Techniques such as memory swapping [49, 6] are employed to offload model parameters or KV caches based on sparsity. Additionally, some systems utilize CPUs to assist GPU operations during model inference or generation [26]. These systems however, still introduce performance degradation.

### 3.3 Benchmarking use cases of CAP analysis

Different MoE systems balance trade-offs among cost, accuracy, and performance differently: some prioritize speed, others hardware efficiency or accuracy. Due to their sparsity and flexible routing, MoE systems exhibit distinct trade-off patterns based on these priorities. For instance, interactive tasks or batch pipelines may favor performance–cost trade-offs over accuracy, while accuracy-critical tasks like retrieval QA or recommendations may tolerate higher cost or latency. Many deployments also face strict cost limits, favoring low-power or budget-friendly setups.

The CAP radar offers a clear view of system trade-offs across cost, accuracy, and performance to support diverse deployment needs. To further illustrate the usage of the CAP Radar, Figure 3 presents two examples running on GSM8K that illustrate these trade-offs in practice.

One important use case is that practitioners often need to choose a MoE system based on their target scenarios. Consider the case where they aim to bound decoding latency while optimizing cost and accuracy. For this, they can use the CAP Radar diagram to profile different candidate systems. The results shown in the left diagram compare SGLang, K-Transformers, and MoE-Infinity based on decoding latency (s/token), purchase cost (USD), and exact match accuracy, all evaluated on NVIDIA A5000 GPUs. SGLang offers the lowest latency and strong accuracy but comes at the highest cost, making it suitable for applications where responsiveness is critical. K-Transformers provides faster decoding than MoE-Infinity at a lower cost but with reduced accuracy, making it a good fit for cost- and speed-driven deployments. MoE-Infinity maintains the same accuracy as SGLang (91.1%) and reduces purchase cost by nearly 60%, but at the expense of a 2.6× increase in latency (0.15 s/token vs. 0.058 s/token). This highlights its trade-off as a viable choice for accuracy-sensitive applications where budget matters more than real-time speed.

Another common use case is that practitioners want to compare quantization against offloading—where the former reduces cost at the expense of accuracy, while the latter preserves accuracy but introduces performance overhead. The CAP Radar diagram aids in this comparison, as shown in the right diagram, which compares MoE-Infinity, SGLang-FP8, and SGLang-AWQ (INT4) on the Qwen3-235B-A22B model, evaluated by power cost (W), decoding throughput (tokens/s), and exact match accuracy on NVIDIA H20 GPUs. MoE-Infinity delivers the highest accuracy and lowest power cost but suffers from low throughput, limiting its use in high-volume workloads. SGLang-FP8 achieves the highest throughput—over 75× faster than MoE-Infinity—with moderate accuracy (92.2%) but at 2.7× higher power cost, making it ideal for batch generation tasks like summarization or embedding extraction. SGLang-AWQ provides a balanced trade-off across all metrics. Depending on deployment goals—energy efficiency, model quality, or throughput—each system offers distinct strengths.

## 4 Sparsity-Aware Performance Metrics

### 4.1 Sparse model bandwidth utilization

When designing a new sparsity-aware MBU, we want this new metric to be generally applicable to all types of MoE models we are aware of by far. These different sparsity patterns of MoE models can be summarized in Figure 2. From this figure, we can derive several requirements for the new MBU. First, the metric must capture the set of experts activated by a batch of input tokens. For example, in case (a), a sparse feed-forward (FF) layer contains three experts, and each token is routed to its top-1 expert. Both tokens are routed to the same expert, so the accessed memory corresponds to a single expert of size $S$. In contrast, case (b) routes the tokens to two distinct experts, doubling the accessed memory to $2S$. Second, the metrics must account for different routing mechanisms, such as shared experts introduced by recent MoE studies [10, 4] (illustrated in Figure 2 (c)).

To meet the above requirements, we define Sparse Memory Bandwidth Utilization (S-MBU) based on the activated size of parameters $S_{\text{activated}}$, rather than using the full model size $S_{\text{model}}$, as follows.

$$\text{S-MBU} = \frac{B_{\text{achieved}}}{B_{\text{peak}}}, \ B_{\text{achieved}} = \frac{S_{\text{activated}} + S_{\text{KV}}}{\text{TPOT}}, \ S_{\text{activated}} = n_{\text{layer}} \times S_{\text{attn}} + \sum_{l=1}^{n_{\text{layer}}} \sum_{i=1}^{n_{\text{expert}}} \mathbb{1}[l, i] \times S_{\text{expert}},$$

(4)

where $\mathbb{1}[l, i]$ is a boolean variable indicates whether the expert indexed $i$ at layer $l$ is used for computation. This guarantees that only the activated parameters are accounted for the accessed memory for each layer $l$. $\mathbb{1}[l, i]$ can be achieved by tracing router outputs.

Besides, dense models are a special case of equation (4) with $n_{\text{expert}} = 1$ and $\forall i, \mathbb{1}[l, i] = 1$. Therefore, our definition is also suitable for model architectures where not all layers are MoE layers, *e.g.*, in Switch Transformers [14]. We further validate S-MBU accuracy on MoE models; detailed results are provided in Appendix A.4.

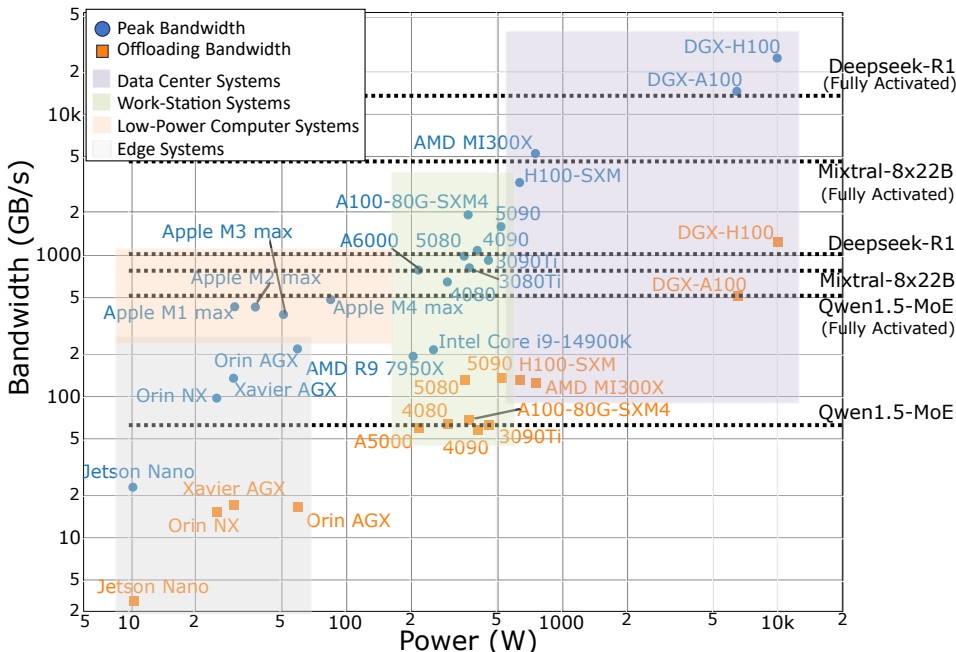

Figure 4: Benchmarking MoE deployment using sparsity-aware performance metrics. Horizontal lines show the minimum bandwidth required for MoE models to meet a decoding latency target, under two scenarios: full activation (large batch size) and minimal activation (batch size = 1). Blue dots represent each device's peak bandwidth and TDP; orange dots indicate reduced bandwidth when DRAM offloading is needed. Devices above the lines satisfy the latency requirement. Systems are grouped by deployment class: edge (e.g., robotics, autonomous driving), low-power devices, workstations, and data centers.

## 4.2 Sparse model FLOPS utilization

We aim to account for the fact that experts are sparsely activated when calculating $F_{\text{token}}$, *i.e.* FLOPs per token. Specifically, in each MoE layer, we account for top-k activated experts with shared experts, denoted $k_{\text{expert}}$, which can be obtained from the model configuration without the need for runtime tracing. Besides, we also account for the router component in each MoE layer, $N_{\text{router}}$. The attention layer remains the same. Consequently, we refine the FLOPs per token calculation as follows:

$$\text{S-MFU} = \left(T_{\text{token}} \times \text{S-}F_{\text{token}}\right)/F_{\text{peak}}, \quad \text{S-}F_{\text{token}} = F_{\text{attn}} + 2N_{\text{router}} + 2k_{\text{expert}}N_{\text{expert}}, \quad (5)$$

where $F_{\text{attn}}$ represents the number of FLOPs needed for the attention module and $N_{\text{expert}}$ represents the number of parameters in the expert module. These values can be derived from the model configuration and accurately calculated based on the easily accessible model structure. Furthermore, since the FLOPs for each matrix multiplication are fixed and deterministic, S-MFU can be ensured with high accuracy. We show the accuracy of S-MFU in Appendix A.5 and the results demonstrate that S-MFU matches profiler-measured S-MFU within 0.05% across all settings, showing it accurately captures MoE compute cost.

## 4.3 Benchmarking use cases of sparsity-aware performance metrics

Our sparsity-aware metrics enable accurate evaluation of AI processors for bottleneck-free MoE deployment. Figure 7 plots peak memory bandwidth against power consumption, covering processors from edge devices to data center systems. Each MoE model is shown with two horizontal lines: one for activation at batch size 1 and another for full activation as batch size increases.

The lines are computed **using our accurate S-MBU metric to represent the actual bandwidth requirements** for deploying the model under both lower-bound conditions (batch size = 1) and upper-bound scenarios (full expert activation). Detailed calculation procedures are provided in Appendix A.7.

For instance, full activation of DeepSeek-R1 requires 18,901 GB/s—a level achievable only on high-end data center hardware like the DGX-H100 (10,200W) using expert parallelism. In contrast, at batch size 1, the bandwidth requirement drops to 1,040 GB/s, making it feasible on consumer GPUs

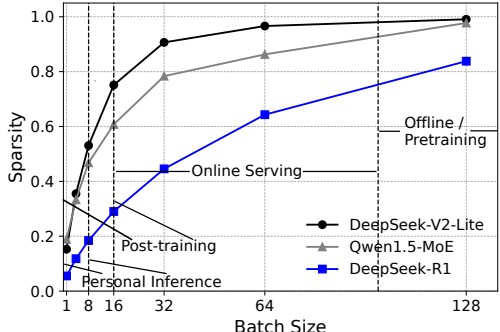
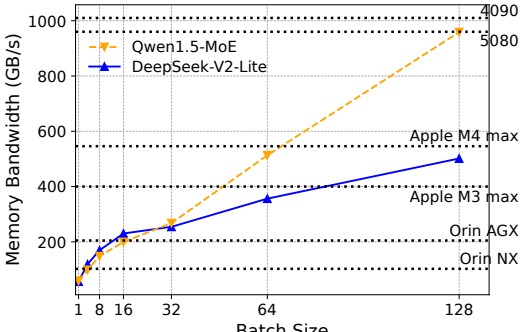

Figure 5: Illustration of how model sparsity varies with batch size, along with the corresponding deployment scenarios on DeepSeek-V2-Lite, Qwen1.5-MoE and DeepSeek-R1

Figure 6: Benchmarking results of MoE-CAP on DeepSeek-V2-Lite and Qwen1.5-MoE, highlighting how their practical bandwidth requirements and hardware choices vary with batch size

such as the RTX 4090 (450W) when paired with efficient offloading strategies (e.g., CA systems). All results assume a TPOT SLO of 0.1s/token.

These findings support the growing market view that MoE models may shift LLM deployment from power-hungry data centers to a broader range of affordable, personal computing platforms.

**Impact of batch size on MoE sparsity and deployment.** In practice, the sparsity of MoE systems is closely influenced by the batch size. Batch size, in turn, is largely determined by the deployment context. For offline batched inference and pretraining, large batch sizes are common, leading to full or near-full activation of experts and thus low sparsity. In contrast, personal devices that support a single user typically operate with batch size 1, meaning high sparsity. In post-training or online inference on shared devices, batch sizes are moderate (ranging from 1 to tens), causing model sparsity to vary dynamically—from high to low—as batch size increases.

To more accurately benchmark performance across scenarios, we analyze how model sparsity evolves with batch size for three MoE models—DeepSeek-V2-Lite, Qwen1.5-MoE, and DeepSeek-R1—and examine the implications for bandwidth requirements and hardware selection.

Figure 5 illustrates the relationship between batch size and model sparsity, based on profiling using our custom benchmarking tool built on the latest vLLM release and evaluated on the MATH [20] dataset. In scenarios such as personal inference and post-training—characterized by small batch sizes (1–16)—all models exhibit high sparsity. At batch size 8, for instance, only 53.05%, 46.79%, and 18.44% of parameters are activated for DeepSeek-V2-Lite, Qwen1.5-MoE, and DeepSeek-R1, respectively. This significantly reduces bandwidth demands, making these models viable for deployment on cost-efficient hardware.

As shown in Figure 6, under a TPOT (time-per-output-token) SLO of 0.25s/token and using S-MBU (see §4.1) for practical bandwidth estimation, we observe that: (i) **Apple M3 Max** meets bandwidth requirements for DeepSeek-V2-Lite at batch sizes of 32; (ii) **NVIDIA Orin AGX**, an edge-class device, supports deployment up to batch size 16; and (iii) **Orin NX**, a lighter variant, remains sufficient for batch sizes up to 4.

These findings highlight that MoE systems can run on various low-power processors, but their performance depends on the deployment scenario—underscoring the need for accurate, sparsity-aware performance metrics to guide hardware selection.

## 5 Benchmark Implementation

**Expert activation profiler.** To evaluate S-MBU accurately, we profile expert activation patterns at a given batch size. We implement lightweight profilers in SGLang and HuggingFace Transformers, inserting probes near the router in each MoE layer to record activations during forward passes. The model runs on representative data until the activation distribution stabilizes. To avoid redundant runs, we store activation sheets for reuse in future evaluations.

**Automated evaluation pipeline.** Following HuggingFace's leaderboard design, we built MoE-CAP as an automated benchmarking tool to evaluate MoE systems across cost, accuracy, and performance. Model and dataset setup, as well as evaluation, are fully automated—users simply provide system and hardware details to run benchmarks. Currently, MoE-CAP supports six widely used MoE-enabled LLM inference frameworks: vLLM, MoE-Infinity, SGLang, K-Transformers, HuggingFace Transformers and Accelerate.

**Dataset support.** We evaluate all models on four representative benchmarks: **MMLU**, **GSM8K**, **MATH**, **Arena-Hard**, and **LongBench**. **MMLU** covers 57 diverse multiple-choice tasks to assess factuality and reasoning across domains. **GSM8K** and **MATH** focus on mathematical reasoning, with problems requiring multi-step solutions and short-form generation at varying difficulty levels. **Arena-Hard** evaluates long-form generation using 500 complex user queries from Chatbot Arena, judged by GPT-4-Turbo. It shows strong agreement with human preferences (89.1%) and better separability than other benchmarks. **LongBench** is a benchmark of long-context questions, making it well-suited for evaluating prefill-heavy workloads. Together, these benchmarks comprehensively test MoE LLMs across multiple-choice, short-form, and long-form generation tasks—covering knowledge, reasoning, and output quality—and are widely used in leading LLM leaderboards [5, 24, 22].

**Model support.** We have currently evaluated the following models: Mixtral-8x7B-Instruct-v0.1, Mixtral-8x22B-Instruct-v0.1, DBRX-Instruct, Qwen1.5-MoE-A2.7B-Chat, DeepSeek-V2-Lite-Chat, Qwen3-30B-A3B, Qwen3-235B-A22B, and DeepSeek-R1, along with their corresponding quantized versions. These MoE models were selected for their diversity in parameter scale and architectural design, as well as for their strong performance. All are widely recognized and adopted across both academic research and industrial applications.

# 6 Insights and Takeaway Messages

MoE-CAP derives key insights that highlight both the potential and challenges of MoE systems:

**MoE systems enable a broader range of devices to perform inference.** Personal machines typically operate in single-user inference environments, where experts are sparsely activated with a small batch size (usually as 1). In this case, many low-power hardware platforms can run large MoE models (with the support of offloading) or smaller versions of MoE models (such as Moonlight-16B-A3B) with reasonable latency and performance. This challenges the conventional belief that large MoE models must run on expensive data center systems, as demonstrated in Figure 7.

**Hybrid computing will become more prevalent.** With emerging matrix acceleration capabilities in processors (e.g., Intel AMX, ARM SME), these processors can significantly assist GPUs in serving MoE models. They also offer greater memory flexibility, making them attractive for MoE deployment. For instance, DRAM on personal workstations can be scaled to TB levels, whereas consumer-grade GPUs are limited to 48GB. As a result, hybrid architectures combining CPUs, host memory, and GPUs will become increasingly common, especially in personal environments where small batches are prevailing.

**MoE systems should be co-designed to align with specific applications and deployment scenarios.** We observe that the sparsity characteristics of MoE models are significantly affected by the applications (e.g., online inference, offline inference, fine-tuning and pre-training) and deployment scenarios (e.g., single-user vs. multi-user, small vs. large batch sizes). These characteristics determine system bottlenecks, which can be addressed by various hardware platforms—some offering significantly lower costs than conventional GPU-only solutions. We anticipate a surge in specialized MoE systems tailored to specific applications and scenarios.

**New benchmarking and design principles are needed for emerging sparse AI systems.** Sparsity is not unique to MoE—it also plays a critical role in emerging sparse AI systems, such as those involving sparse KV-cache architectures (common in long-context reasoning) and module-based agentic LLM workflows. In these cases, sparsity directly influences system bottlenecks, highlighting the need for new system benchmark, profiling and design principles that inherently account for cost, performance, and accuracy. With these principles, a wide range of sparsity-aware system optimizations can be unlocked. Without them, sparse AI systems risk falling short of their cost-saving potential and performance goals.

## Acknowledgements

Luo Mai, Edoardo Ponti, and Yinsicheng Jiang are supported by the Advanced Research and Invention Agency (ARIA)'s grant "*Scaling Compute: AI at 1/1000th the cost. Technical Area 4 Benchmarking*".

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

# A More Details of MoE-CAP

## A.1 MoE-CAP Limitation and Future Work

In this section, we discuss the current limit and future work for MoE-CAP.

**Limitation.** The current version of MoE-CAP focuses exclusively on inference tasks. It does not yet address other important deployment scenarios such as post-training and pre-training. Additionally, broader evaluation across a wider range of MoE models and systems is needed to ensure comprehensive coverage and generalizability.

**Future Work.** For future work, we will investigate MoE-CAP within training settings, instrumenting training tasks to evaluate its capabilities.

**System Selection Map for broader deployment benchmarking.** As part of future work, we plan to expand MoE-CAP with a System Selection Map that tracks the best-in-class AI software and hardware systems across diverse deployment scenarios, including inference, post-training, and pre-training. Building on recent advances in systems like vLLM, SGLang, MoE-Infinity, Unsloth, Axolotl, and TorchTitan, this map will benchmark SOTA models—identified in the Model Selection Map—on a range of platforms from single-node GPUs to multi-node clusters and heterogeneous architectures. This will enable scenario-specific benchmarking (e.g., short-context vs. long-context inference) to guide optimal system-model pairing.

To accelerate end-to-end benchmarking, we package the system as a prebuilt Docker image and expose CAP analysis as a service (Figure 7). Datasets and models are mounted from our storage at runtime, and the container runs a command that launches the FastAPI-based CAP analyzer. During inference, the analyzer is invoked (`Post /cap-profiler`) on every forward pass to collect data and compute CAP metrics. When all user requests finish, the service assembles the final report and exposes it via a pull endpoint (`Get /cap-results`).

**Toward real-world deployment scenarios.** We also aim to evaluate MoE systems in diverse cloud deployment settings, including serverless model endpoints [18], elastic infrastructure [43, 45, 34], and spot-instance pricing environments [35, 36, 42]. This will enable more realistic assessments of cost-performance trade-offs under variable resource conditions.

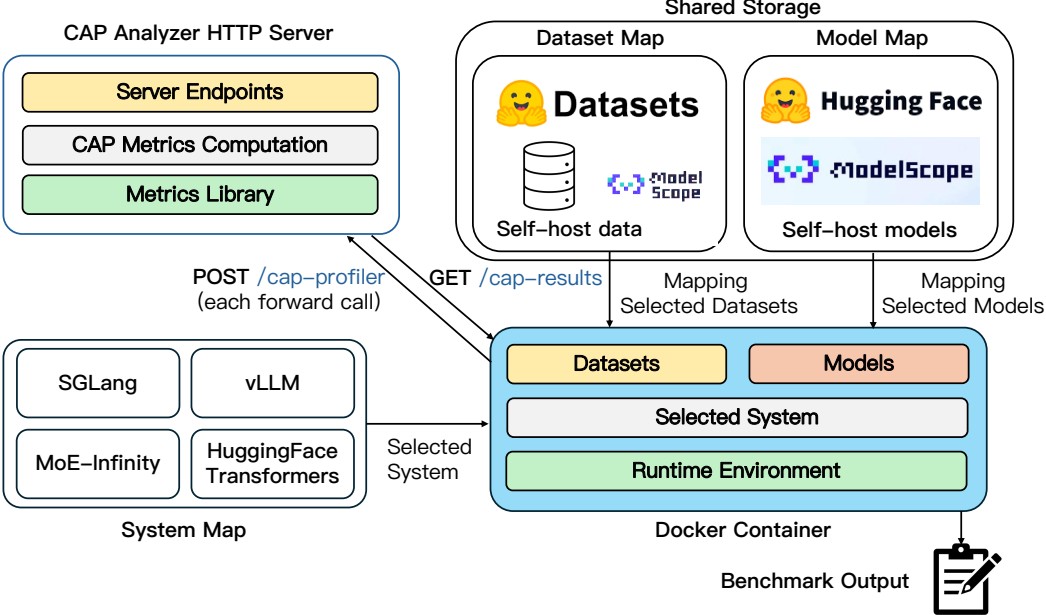

Figure 7: MoE-CAP evaluation pipeline implementation

**A.2  Cost model use cases**

We have considered several use cases after introducing new cost models as part of our benchmark. For instance, in GPU-only deployments, the specifications for DRAM, CPU, and SSD are typically not defined. Cloud providers often scale the computational power in line with GPU capabilities, offering four times the DRAM capacity (e.g., 2TB DDR5) relative to GPU memory (e.g., 8XA100-80GB), and pair it with the latest CPUs (e.g., AMD 9004 series). This hardware combination results in an approximate cost of $176,000 per server. Opting for a less powerful CPU and reduced memory capacity in GPU-only workloads can yield a saving of $20,000 per server in hardware costs.

Additionally, for MoE systems that enable offloading, it is crucial to account for the CPU's energy consumption and the energy used in communication between the CPU and GPU. For instance, the AMD 777X has a peak power consumption of 280W, while the A6000 Ada peaks at 300W. Relying solely on GPU energy assessments might lead to overly optimistic forecasts for energy savings, as the CPU's power consumption can be as significant as another GPU.

**A.3  S-MBU and S-MFU on dynamic batching**

Dynamic batching introduces variability in execution, as the number of forward passes required to process a given query queue depends on sequence length distribution, dataset characteristics, and hardware constraints. For instance, a batch of 128 queries may be executed in 2 or 4 separate forward passes depending on how sequences are packed.

To accurately capture system behavior, our probing mechanism instruments both the prefill and decoding stages. On each forward pass, we record the S-MBU (as defined in Equation 6), the actual batch size, and the per-layer expert activation patterns, including token-level routing information. This comprehensive data allows us to compute the data movement through activated experts for every pass. The formulation ensures that S-MBU remains valid and comparable across workloads with heterogeneous sequence lengths. By grounding the metric in actual routing behavior and hardware-observed throughput, it provides a reliable indicator of sparsity-induced memory bottlenecks, even under realistic and variable deployment scenarios. In addition to S-MBU, S-MFU is based solely on token-level throughput ($T_{\text{token}}$). Similar to S-MBU, we get the token throughput of each batch and then calculate the S-MFU.

$$\text{S-MBU} = \frac{\dfrac{\sum_{\text{forward}} \left(S_{\text{activated}} + S_{\text{KV}}\right)}{\sum_{\text{forward}} \text{Latency}}}{\text{Hardware Memory Bandwidth}} \tag{6}$$

**A.4  S-MBU accuracy on MoE models**

We show that our S-MBU definition can capture novel MoE architectures. Since S-MBU accounts for actual expert activation in MoE models, we evaluate its accuracy against the standard (vanilla) MBU definition.

Figure 8 shows the average accessed memory for a specific Transformer layer in the Mixtral-8x7B model using the GSM8K dataset, with batch sizes ranging from 1 to 64 and running on A100-PCIe-80G. At batch size 1, each layer activates only the top 2 experts. As batch size increases, more experts are accessed—but the total number of activated experts does not grow linearly, since tokens may share experts.

The results demonstrate that vanilla MBU significantly overestimates memory usage, by over 260%, due to its failure to account for selective expert activation. In contrast, S-MBU closely matches actual memory usage, with less than 1% error compared to profiles based on HuggingFace Transformers.

We also observe layer-wise variation in expert activation as batch size grows. For example, at batch size 64, only about half of the experts are activated in the first layer, whereas in deeper layers (e.g., layers 16 and 32), nearly all experts are used. S-MBU accurately reflects the trend of increased memory access with larger batches. However, when most experts are activated, both MBU and S-MBU still underestimate total memory due to excluding intermediate states (e.g., hidden activations), which scale linearly with batch size. We show further experiments on Qwen to demonstrate the accuracy of S-MBU. Qwen has a distinct model architecture in MoE layers. Each MoE layer has 64 experts where the first 4 experts are shared by all tokens. In other words, each token, apart from

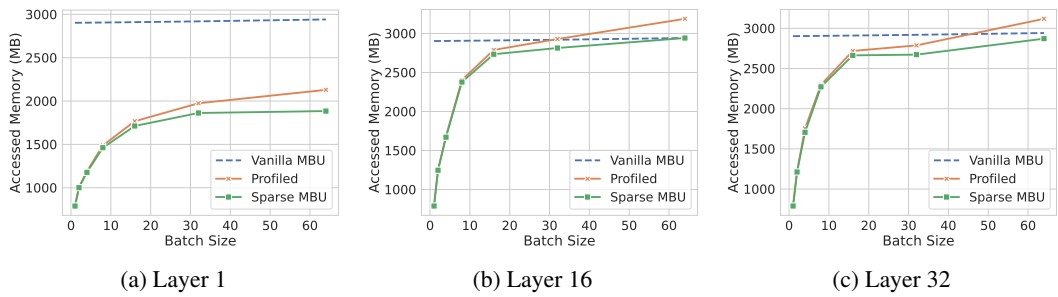

(a) Layer 1           (b) Layer 16           (c) Layer 32

Figure 8: Correctness evaluation of the vanilla MBU, our S-MBU and actual MBU (through profiling).

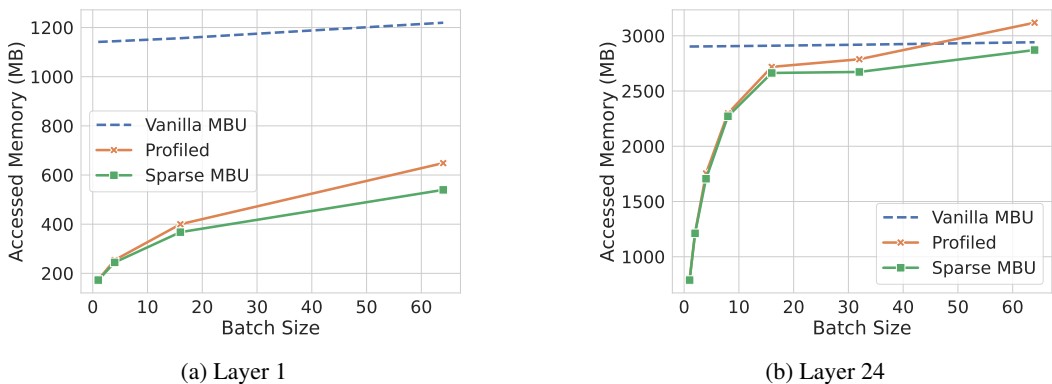

(a) Layer 1                  (b) Layer 24

Figure 9: Accuracy evaluation of the vanilla MBU, our S-MBU and actual MBU (through profiling).

top-$4$ experts, is also processed by 4 shared experts. Therefore, we have $\mathbb{1}[l, i] = 1, i = 1, 2, 3, 4$ and $\forall l \in [1, n_{\text{layer}}]$. For $i > 4$, $\mathbb{1}[l, i]$ can still be traced as other MoE models.

We profiled the memory access of one decoder layer (the first and last layer) of Qwen model using HuggingFace Transformers and show the comparison of vanilla MBU and Sparse MBU in Figure 9.

In both layers, with batch size increasing, the memory footprint increases sub-linearly. This is because tokens can activate the same experts. Even with batch size $= 64$, still not all experts are activated. If we do not consider this sparse activation (*i.e.* Vanilla MBU), the overestimation is even more severe. And similar to Mixtral models, we also observe that there are more overlap in expert activation of batched tokens in layer 1 (*i.e.* the first layer) than in layer 24 (*i.e.* the last layer). Different from Mixtral models, only $4/60$ non-shared experts are selected for each token. Therefore, even with batch size $= 64$, not all experts are accessed. We do not evaluate MBU in larger batch size because MBU is only meaningful in memory bandwidth-bounded scenarios (*i.e.* small batch sizes).

This experiment demonstrates that S-MBU definition can be extended to cooperate novel architectures and the calculated memory access is accurate too.

### A.5 S-MFU accuracy on MoE models

We validate S-MFU on Mixtral-8×7B and DeepSeek-V2-Lite over batch sizes 1–32. As shown in Table 2, our analytic S-MFU matches profiler-measured S-MFU within 0.05% across all settings, showing it accurately captures MoE compute cost.

### A.6 S-MBU on multi-node inference

To ensure the real-world reliability, accuracy, and practical value of our proposed metrics, we have maintained close collaboration with an industrial partner to validate the metrics under realistic deployment scenarios, including multi-node inference.

We present results evaluating the accuracy of S-MBU in a production-like setup, where our collaborator deployed the SGLang serving framework on a two-node cluster—each node equipped with

Table 2: S-MFU Accuracy

| Model | Batch size | Profiled S-MFU (%) | Ours (%) |
|---|---|---|---|
| Mixtral-8x7B | 1 | 0.08 | 0.06 |
| | 4 | 0.19 | 0.18 |
| | 8 | 0.31 | 0.29 |
| | 16 | 0.50 | 0.48 |
| | 32 | 0.85 | 0.80 |
| DeepSeek-V2-Lite | 1 | 0.01 | 0.01 |
| | 4 | 0.04 | 0.03 |
| | 8 | 0.05 | 0.05 |
| | 16 | 0.07 | 0.06 |
| | 32 | 0.07 | 0.07 |

8×NVIDIA H20 GPUs and connected via 400 GB/s InfiniBand—running the DeepSeek-R1 model on the LongBench dataset. In this setting, analytically computed S-MBU values were compared against actual communication utilization profiled using torch.profiler.

As shown in the Table 3, the computed S-MBU values closely match the profiled results across all batch sizes, with deltas consistently below 1%. This alignment supports the robustness of S-MBU in capturing communication efficiency under practical multi-node scenarios.

Table 3: Performance metrics across different batch sizes.

| Batch Size | S-MBU (%) | Profiled S-MBU (%) | $\Delta$ (%) |
|---|---|---|---|
| 1 | 1.67 | 1.88 | 0.21 |
| 16 | 7.91 | 8.15 | 0.24 |
| 32 | 10.08 | 10.38 | 0.30 |
| 64 | 17.33 | 17.75 | 0.42 |
| 128 | 33.12 | 33.91 | 0.79 |

### A.7 Practical bandwidth and compute requirement calculation

We calculated theoretical performance through existing tools like LLM-Analysis [29] and LLM-Viewer [53]. In practice, MoE systems often suffer inefficiencies due to redundant data transfers or calculations, resulting in performance losses. Consequently, the calculated theoretical bandwidth or OPS may not meet the required application performance (e.g., throughput). To determine the practical hardware requirements for a given set of inputs, it is essential to consider hardware utilization. Thus, the practical performance requirement is defined as:

$$\text{Practical Bandwidth} = \frac{\text{Theoretical Bandwidth}}{\text{S-MBU}} \tag{7}$$

$$\text{Practical OPS} = \frac{\text{Theoretical OPS}}{\text{S-MFU}} \tag{8}$$

Here, S-MBU and S-MFU (defined in the next subsection) represent the actual hardware utilization for MoE models.

To illustrate, suppose the theoretical bandwidth requirement is 400 GB/s. If the hardware achieves only 50% bandwidth efficiency during model decoding, the practical bandwidth requirement must be doubled to 800 GB/s ($\frac{400\,\text{GB/s}}{50\%}$) in order to meet the same throughput or latency needs.

### A.8 Results on prefill stage

MoE-CAP profiler already records prefill throughput and time-to-first-token (TTFT). Some results are shown in the table 4. We are expanding our benchmark to include long-context workloads such

as LongBench. Early experiments demonstrate that S-MBU and S-MFU effectively characterize decoding performance under extended sequences. These results show that full expert activation during prefill increases latency and can alter the optimal MoE configuration.

Table 4: Benchmark results on prefill stage for different models and methods.

| Model | Method | Benchmark | Batch Size | TTFT (s) | Prefill Throughput (T/s) | Device |
|---|---|---|---|---|---|---|
| Qwen1.5-MoE-A2.7B-Chat | hf-chat | GSM8K | 16 | 0.06 | 4166.67 | 1xA100-80G-SXM4 |
| Qwen1.5-MoE-A2.7B-Chat | hf-chat | Arena Hard | 16 | 0.06 | 4132.63 | 1xA100-80G-SXM4 |
| Qwen1.5-MoE-A2.7B-Chat | hf-chat | LongBench | 1 | 23.23 | 561.82 | 4xRTX A6000 |
| Qwen3-30B-A3B | sglang | GSM8K | auto | 0.01 | 21844.81 | 4xRTX A6000 |
| Qwen3-30B-A3B | sglang | Arena Hard | auto | 0.02 | 6993.28 | 4xRTX A6000 |
| Qwen3-30B-A3B | sglang | LongBench | auto | 2.3 | 5652.17 | 4xRTX A6000 |
| DBRX-Instruct | vllm_moe | GSM8K | auto | 0.08 | 3125.00 | 8xA100-80G-PCIe |
| DBRX-Instruct | vllm_moe | Arena Hard | auto | 0.12 | 2083.33 | 8xA100-80G-PCIe |
| DBRX-Instruct | hf-chat | GSM8K | 16 | 0.40 | 625.00 | 4xA100-80G-PCIe |
| DBRX-Instruct | hf-chat | Arena Hard | 8 | 0.29 | 862.07 | 4xA100-80G-PCIe |
| Mixtral-8x22B-Instruct | vllm_moe | GSM8K | auto | 0.14 | 1776.54 | 8xA100-80G-PCIe |
| Mixtral-8x22B-Instruct | vllm_moe | Arena Hard | auto | 0.14 | 1785.71 | 8xA100-80G-PCIe |
| Mixtral-8x22B-Instruct | hf-chat | GSM8K | 16 | 0.43 | 581.40 | 4xA100-80G-PCIe |
| Mixtral-8x22B-Instruct | hf-chat | Arena Hard | 8 | 0.27 | 925.93 | 4xA100-80G-PCIe |
| Mixtral-8x7B-Instruct | vllm_moe | GSM8K | auto | 0.01 | 18932.23 | 8xA100-80G-SXM4 |
| Mixtral-8x7B-Instruct | vllm_moe | Arena Hard | auto | 0.05 | 5024.03 | 8xA100-80G-SXM4 |
| Mixtral-8x7B-Instruct | hf-chat | GSM8K | 64 | 0.15 | 1666.67 | 2xA100-80G-SXM4 |
| Mixtral-8x7B-Instruct | hf-chat | Arena Hard | 16 | 0.17 | 1470.59 | 2xA100-80G-SXM4 |

## A.9 Profiling overhead on latency and metric error

We recorded the overhead before and after adding our runtime profiling tools. Our expert-activation profiler uses lightweight tensor operations that remain compatible with CUDA graph compilation to minimize disruption. As shown in table 5, in our benchmarks—comparing vLLM inference with and without the profiler—we observed a maximum 2.7% overhead of just 8 ms in Time-To-First-Token (TTFT) and 2.2% overhead - 4ms in Tokens-Per-Output-Token (TPOT), confirming that the profiling imposes a negligible performance penalty.

We also recorded statistical measures including standard deviations. As shown in the table 6, the standard deviations for key metrics—including decoding S-MFU, S-MBU, prefill latency, and model accuracy—remain consistently low across all experimental settings. These small deviations do not affect the core insights or conclusions of our comparative analysis.

Table 5: Performance comparison before and after optimization.

| Model | Framework | Hardware | Batch Size | TTFT (Before) | TTFT (After) | ΔTTFT | TPOT (Before) | TPOT (After) | ΔTPOT |
|---|---|---|---|---|---|---|---|---|---|
| Qwen3-235B-A22B | SGLang | 8× H20 | auto | 0.067 | 0.071 | +0.004 | 0.019 | 0.021 | +0.002 |
| Qwen3-30B-A3B | SGLang | 4× A6000 | auto | 0.020 | 0.024 | +0.004 | 0.059 | 0.063 | +0.004 |
| Mistral-8x7B | Huggingface Transformers | 2× A100-SXM4 | 16 | 0.173 | 0.176 | +0.006 | 0.179 | 0.183 | +0.004 |
| DBRX | Huggingface Transformers | 4× A100-PCIe | 8 | 0.290 | 0.298 | +0.008 | 0.286 | 0.286 | +0.000 |

Table 6: Model evaluation results under different MoE systems and hardware with standard errors.

| Model | Eval Type | Exact Match | Exact Match StdErr | Prefill Time (s) | Prefill Time StdErr | Decoding Throughput (T/s) | Decoding Throughput StdErr | Decoding MFU | Decoding MFU StdErr | Decoding MBU | Decoding MBU StdErr |
|---|---|---|---|---|---|---|---|---|---|---|---|
| Qwen1.5-MoE (HF, 1×A100 PCIe) | Best Match | **0.4769** | 0.01371 | 0.05655 | 0.00023 | 41.3253 | 0.02629 | 0.00560 | 0.000037 | 0.03796 | 0.000022 |
| Qwen3-30B-A3B (SGLang, 4×A6000) | Best Match | **0.9014** | 0.00821 | 0.00582 | 0.00003 | 1343.1684 | 2.81220 | 0.00208 | 0.000004 | 0.18099 | 0.000460 |
| Qwen3-235B-A22B (SGLang, 8×H20) | Best Match | **0.8961** | 0.00840 | 0.06550 | 0.00086 | 1792.0811 | 4.87791 | 0.01616 | 0.000043 | 0.14942 | 0.000397 |

## A.10 Multi-constraint decision matrix of CAP analysis

We distill CAP analysis into deployment heuristics or templates for the CAP radar plots in section 3.3 to provide clear guidance for deployments under different use cases. As shown in table 7, the matrix

is built by actionable "if–then" rules and it clearly guides users to choose suitable MoE systems under certain constraints.

Table 7: Extended Multi-Constraint Decision Matrix of CAP Analysis.

| Hardware Tier | Batch Size | Primary Constraint | Secondary Constraint | Recommended System | Recommended Configuration | Recommendation Reason | Example Use Case |
|---|---|---|---|---|---|---|---|
| Workstation GPU (A5000) | 8+ | Performance (Latency) | Accuracy | SGLang/vLLM | FP16 | Original accuracy with lowest latency | Chain-of-Thought inference |
| Workstation GPU (A5000) | 1-8 | Cost | Latency | K-Transformers | Quantization | Low cost and moderate speed | Chatbot |
| Workstation GPU (A5000) | 1-8 | Accuracy | Cost | MoE-Infinity | Expert offloading | Original accuracy with low cost | Model benchmarking |
| Datacentre GPU (H20) | 1-16 | Accuracy | Power Cost | MoE-Infinity | Expert offloading | Original accuracy with low power cost | Model benchmarking |
| Datacentre GPU (H20) | 16+ | Performance (Throughput) | Power Cost | SGLang/vLLM-FP8 | Mixed precision | High throughput with accepetable accuracy | Batch document retrieval |
| Datacentre GPU (H20) | 16+ | Performance (Throughput) | Accuracy | SGLang/vLLM | FP16 | High throughput with original accuracy | Offline batch processing |

## A.11 Stress test with batch-size spikes on MoE systems

To emulate sudden batch-size spikes, we replayed Microsoft Azure request traces and parameterized inter-arrival times with a Poisson distribution, following established serving workloads. We compared two deployment strategies for Qwen3-30B-A3B on A6000 GPUs: MoE-Infinity with a fixed batch and SGLang with adaptive, continuous batching. The result is shown in table 8. Under rising request rates, SGLang achieved a higher S-MBU, peaking at 54.5% versus 16.7% for MoE-Infinity, demonstrating that continuous batching better maximizes S-MBU under load. However, as request volume and sequence length increased, token eviction in SGLang's PA system led to a sharp S-MBU decline under saturation. In contrast, MoE-Infinity's offload and fixed-batch approach yielded steadier, though lower, utilization. These results show that PA systems are more sensitive to such spikes and may require additional redundancy to maintain performance.

Table 8: MoE system performance across different time intervals.

| MoE System | 30s | 60s | 90s | 120s | 150s | 180s | 210s |
|---|---|---|---|---|---|---|---|
| SGLang | 35.62% | 42.44% | 50.01% | 54.52% | 29.78% | 27.77% | 11.64% |
| MoE-Infinity | 22.95% | 15.25% | 16.73% | 14.25% | 15.95% | 15.12% | 13.97% |

