# OpenReview forum: "MoE-CAP: Benchmarking Cost, Accuracy and Performance of Sparse Mixture-of-Experts Systems"
_NeurIPS.cc/2025/Datasets_and_Benchmarks_Track — NeurIPS 2025 Datasets and Benchmarks Track poster_

### Official Review · Reviewer_c2pe · 2025-06-30

**Rating:** 5
**Confidence:** 3

**Summary:**

This paper presents MoE-CAP, a new benchmark designed to evaluate sparse Mixture-of-Experts (MoE) systems based on three essential dimensions: Cost, Accuracy, and Performance (CAP). The authors introduce new sparsity-aware metrics, Sparse Memory Bandwidth Utilization (S-MBU) and Sparse Model FLOPS Utilization (S-MFU), to address the inadequacy of existing benchmarks that fail to account for sparse expert activation. In addition, they propose the CAP radar diagram for visualizing trade-offs among the three dimensions and offers an automated benchmarking pipeline supporting multiple LLMs and hardware configurations.

**Additional Feedback:**

- **Accuracy of S-MFU**: While the paper thoroughly justifies the Sparse Model FLOPS Utilization (S-MFU) metric conceptually, it lacks empirical evaluation or validation (similar to what was provided for S-MBU in Appendix A.2). Including experimental comparisons (e.g., against profiled FLOPS or existing baselines) would strengthen confidence in S-MFU’s practical accuracy.
- **Model support clarification**: The paper claims support for Qwen3 and DeepSeek-R1 in the benchmarking pipeline, but these models do not appear on the HuggingFace Open LLM Leaderboard. It would be helpful to clarify whether they are omitted due to evaluation constraints, licensing, or if they are evaluated solely in the authors’ custom leaderboard.
- **Minors**
    - Clarify all notations introduced in Section 2.1
    - Typos in Eq. (2): $F_{\text{attn}} \to N_{\text{attn}}$
    - Improve the readability of Figure 4
    - In Figure 5, there are two arrows pointing to post-training

**Dataset Code Accessibility:**

Yes

**Dataset Code Comments:**

The authors provide a public GitHub repository (https://github.com/sparse-generative-ai/open-moe-llm-leaderboard) containing the benchmarking code and datasets. The automated evaluation pipeline, expert activation profilers, and model configurations are all described in Section 5 and Appendix A. The setup appears reproducible and well-documented.

**Ethical Comments:**

The paper uses only public datasets and focuses on system-level benchmarking of MoE models, with no data collection, annotation, or user interaction involved. It raises no significant concerns around privacy, fairness, or representativeness. The authors briefly discuss environmental efficiency via reduced power needs for certain deployment strategies (e.g., offloading, quantization).

**Ethical Considerations:**

No, there are no or only very minor ethics concerns

**Final Justification:**

The authors have addressed my concerns during the rebuttal. I keep my original score.

**Limitations Weaknesses:**

The first two limitations are acknowledged by the authors in Appendix A.1. Notably, the second—concerning the omission of the prefilling phase—is especially critical for accurately capturing the real-world cost, accuracy, and performance of MoE systems. Are there any (even preliminary) ideas from the authors on how to address this gap in future versions of the benchmark?

- **Limited to inference**: As acknowledged in Appendix A.1, MoE-CAP focuses solely on inference scenarios. Pre-training and post-training workloads which also significantly impact cost and performanceare not currently supported.
- **Pre-fill latency omitted**: Many LLM workloads experience significant latency during the pre-filling phase, especially for long-context inputs. This aspect is absent in the current benchmark, potentially underrepresenting real-world deployment costs.
- **Sparse attention overhead not evaluated**: While the paper mentions the quadratic cost of attention (Section 3.1), it doesn’t benchmark or simulate the added benefit/cost of sparse attention alongside MoE layers, missing an opportunity for holistic sparsity-aware modeling.

**Strengths Contributions:**

- **Timely and impactful contribution**: As MoE-based LLMs become more prevalent (e.g., DeepSeek, Qwen, Mixtral), benchmarking tools that consider sparsity and heterogeneous deployment become increasingly critical.
- **Novel metrics (S-MBU and S-MFU)**: The proposed metrics correct the overestimations in traditional MBU/MFU by considering selective expert activation (see Figure 2, page 3), leading to more accurate resource estimations and better hardware provisioning.
- **CAP Radar Diagram**: This visualization (Figure 3) effectively highlights trade-offs between cost, accuracy, and performance, helping practitioners select MoE systems suited to specific use cass.
- **Comprehensive benchmarking pipeline**: The authors support a wide range of MoE models (e.g., Mixtral, DBRX, DeepSeek-R1) and systems (vLLM, MoE-Infinity, etc.), and benchmark them on multiple datasets (GSM8K, MMLU, MATH, Arena-Hard). This broad coverage increases the benchmark’s utility and reproducibility.
- **Insights for deployment**: The analysis shows how MoE models, when sparsely activated (e.g., batch size = 1), can be run on low-power consumer hardware like RTX 4090 or Apple M-series chips, challenging the assumption that LLMs require data center-grade GPUs (Figure 4, page 7).
- **Well-organized and readable**: The paper is clearly structured, with detailed diagrams, tables, and formulas.

---

> ### Author Rebuttal · Authors · 2025-07-31
>
> ## Response to Limited to inference: As acknowledged in Appendix A.1, MoE-CAP focuses solely on inference scenarios. Pre-training and post-training workloads which also significantly impact cost and performanceare not currently supported.
>
> We thank the reviewer for this suggestion. Our MoE-CAP profiler already records prefill throughput and time-to-first-token (TTFT), and we focused the paper on the decoding phase because it exhibits pronounced variation in expert activation patterns, making it most sensitive to sparsity-aware metrics. In the revised manuscript we will include prefill throughput and TTFT alongside decoding results.
>
> We are expanding our benchmark to include long-context workloads such as LongBench. Early experiments demonstrate that S-MBU and S-MFU effectively characterize decoding performance under extended sequences. These results show that full expert activation during prefill increases latency and can alter the optimal MoE configuration. We will report these findings and incorporate additional datasets in the final version.
>
> Although MoE-CAP currently emphasizes inference due to its diverse sparsity levels, deployment scenarios, and hardware platforms, we are collaborating with AI practitioners to evaluate post-training stages—particularly reinforcement learning fine-tuning with parallel rollouts, which introduce varied activation patterns. Pre-training workloads generally activate all experts and rely on a narrow set of hardware; we intend to investigate sparsity’s role in pre-training in future work.
> | Model                   | Method         | Benchmark   | Batch Size | Prefill Latency (s) | Prefill Throughput (T/s) | Device |
> |-------------------------|----------------|-------------|------------|----------------------|---------------------------|------|
> | Qwen1.5-MoE-A2.7B-Chat  | hf-chat        | GSM8K       | 16         | 0.06                 | 4166.67                   | 1xA100-80G-SXM4 |
> | Qwen1.5-MoE-A2.7B-Chat  | hf-chat        | Arena Hard  | 16         | 0.06                 | 4132.63                   | 1xA100-80G-SXM4 |
> | Qwen3-30B-A3B           | sglang         | GSM8K       | auto       | 0.01                 | 21844.81                  | 4xRTX A6000 |
> | Qwen3-30B-A3B           | sglang         | Arena Hard  | auto       | 0.02                 | 6993.28                   | 4xRTX A6000 |
> | DBRX-Instruct           | vllm_moe       | GSM8K       | auto       | 0.08                 | 3125.00                   | 8xA100-80G-PCIe |
> | DBRX-Instruct           | vllm_moe       | Arena Hard  | auto       | 0.12                 | 2083.33                   | 8xA100-80G-PCIe |
> | DBRX-Instruct           | hf-chat        | GSM8K       | 16         | 0.40                 | 625.00                    | 4xA100-80G-PCIe |
>
> ## Response to Accuracy of S-MFU:
> We appreciate the suggestion. We have benchmarked several MoE models across varying batch sizes and compared our S-MFU calculations against FLOPs utilization reported by the DeepSpeed FLOPs Profiler. This empirical validation confirms the accuracy of S-MFU, and we will include these results in the revised manuscript.
> ## Configuration
> - **Models**: Mixtral-8x7B, DeepSeek-V2-Lite
> - **Batch Sizes**: 1, 4, 8, 16, 32
> - **Dataset**: GSM8K
> - **Hardware**: 8 x NVIDIA RTX A5000 (24GB each)
> - **Total Peak Performance**: 520 TFLOPS (FP16)
>
> | Model            | Batch | Profiled S-MFU (%) | Ours (%) |
> |------------------|-------|----------------|-------|
> | **Mixtral-8x7B** | 1     | 0.08           | 0.06  |
> |                  | 4     | 0.19           | 0.18  |
> |                  | 8     | 0.31           | 0.29  |
> |                  | 16    | 0.50           | 0.48  |
> |                  | 32    | 0.85           | 0.80  |
> | **DeepSeek-V2-Lite** | 1  | 0.01           | 0.01  |
> |                  | 4     | 0.04           | 0.03  |
> |                  | 8     | 0.05           | 0.05  |
> |                  | 16    | 0.07           | 0.06  |
> |                  | 32    | 0.07           | 0.07  |
>
> ## Response to Model support clarification:
> Thank you for pointing this out. We evaluated on our custom leaderboard - Open-MoE-LLM-Leaderboard on Huggingface.
>
> ## Response to Pre-fill latency omitted:
> We have answered the same question as Reviewer 638L.
>
> ## Response to Sparse attention overhead not evaluated:
>
> While our paper focuses on MoE layers, the proposed S-MBU and S-MFU metrics are built on a novel foundational methodology: leveraging runtime profiling instead of static model configuration analysis (as in llm-analysis), which is essential for capturing dynamic sparsity. This approach naturally generalizes to other sparse mechanisms—for example, SpargeAttention (ICML '25), which computes sparse attention masks within the FlashAttention kernel via an online algorithm. In such scenarios, static analysis fails to accurately reflect the true compute and memory access costs. Our profiling-based method enables precise measurement of runtime behavior by inserting lightweight counters—demonstrated in our MoE expert activation case—which in turn makes it straightforward to extend the methodology to sparse attention, allowing for a more holistic evaluation of sparsity in modern AI systems.
>
> ## Response to Minors:
> Thank you for pointing out these minor issues.
> In Section 2.1, we will clarify all introduced notations. Specifically:
> - $S_{kv}$ and $S_{model}$ denote size of KV cache and model weights, respectively.
> - $B_{achieved}$ refers to the achieved memory bandwidth per forward pass.
> - **TPOT** stands for *Time Per Output Token*, representing decoding latency.
> - $B_{peak}$ is the peak bandwidth of the underlying hardware.
> - $T_{token}$ is the token throughput.
> - $F_{token}$ represents the FLOPs required per token.
> - $F_{peak}$ indicates the peak FLOPs capacity of the hardware.
> Regarding Eq. (2), we acknowledge the typo—$N_{attn}$ should be $F_{attn}$, as the equation is based on FLOPs rather than parameter count. Using $N_{attn}$ would not generalize to sparse attention, so this will be corrected in the next version.
> We also appreciate your comments on the visual clarity of Figure 4 and the redundant arrows in Figure 5. These will be revised to improve readability and visual accuracy in the updated manuscript.

---

> > ### Comment · Reviewer_c2pe · 2025-08-02
> > **Response to Rebuttal**
> >
> > I appreciate the authors’ detailed responses to my concerns. I will maintain my original score.

---

### Official Review · Reviewer_aN5G · 2025-07-01

**Rating:** 4
**Confidence:** 3

**Summary:**

This paper proposes MoE-CAP, a practical benchmarking framework to evaluate sparse Mixture-of-Experts (MoE) systems across three deployment-critical dimensions: Cost, Accuracy, and Performance. By introducing sparsity-aware metrics and a categorization of MoE trade-off types, it provides actionable tools for developers and researchers. The framework is supported by a broad and automated benchmark covering multiple inference frameworks, models, and datasets.

**Additional Feedback:**

1. Provide a more formal and unified cost accounting model, including assumptions (e.g., pricing sources, amortization of capital costs, power-to-cost conversion).
2. Distill CAP analysis into deployment heuristics or templates (e.g., per use case, batch size, hardware tier).
3. Consider including a synthetic “stress test” evaluation to show how different MoE strategies behave under extreme or adversarial deployment conditions (e.g., sudden batch size spike or memory throttling).

**Dataset Code Accessibility:**

NA; not applicable to this submission (e.g., no new dataset, benchmark, code, or data provided)

**Ethical Considerations:**

No, there are no or only very minor ethics concerns

**Final Justification:**

Given the authors’ thorough rebuttal and the clarified contributions—particularly the practical impact of the benchmarking framework, deployment heuristics, and stress-tested evaluation—I believe the paper now meets the bar for acceptance and have accordingly raised my score to a borderline accept.

**Limitations Weaknesses:**

1. The newly introduced metrics, while practically useful, are largely structural reinterpretations of existing MBU/MFU rather than algorithmically novel. This isn't inherently problematic—simplicity can be powerful—but it does limit the technical depth of the contribution. A more rigorous derivation or formal treatment of sparsity modeling (e.g., accounting for router dynamics under stochastic token dispatch) might enhance the theoretical foundation.
2. The cost model presented (CAP–Cost) appropriately expands beyond GPU-only accounting to include heterogeneous resources. However, it does not appear to offer a unified formula or cost normalization scheme across devices (e.g., how DRAM+SSD cost is quantified relative to HBM). This may affect reproducibility and comparability, especially in cross-device evaluations (§3, §4.3).
3. While the paper offers interesting system-level observations (e.g., low-batch MoEs can run on edge hardware; hybrid CPU+GPU is emerging), these are not yet distilled into concrete design guidelines or generalized deployment recipes. It would be helpful to include a table or summary highlighting actionable “if–then” rules—for example, “if you prioritize latency within power constraints, prefer quantized PA systems like X; if you prioritize accuracy under tight budgets, use offloading-based CA systems like Y.”
4. As acknowledged in Appendix A.1, the current benchmark focuses on decoding-only inference, which limits its applicability to training-time tuning, long-context prefilling, or multi-user serverless environments. This weakens the generality of the CAP radar diagram in such scenarios, especially given that real deployments increasingly rely on amortized cost analysis over full request lifecycles.

**Strengths Contributions:**

1. As the adoption of sparse MoE architectures accelerates in production-scale LLMs (e.g., Mixtral, DBRX, Grok-1), practical deployment challenges around efficiency and cost-performance trade-offs have emerged as critical bottlenecks. This paper directly targets this unmet need by providing a unified benchmarking framework tailored to MoE systems. Existing LLM benchmarks either ignore sparsity or assume dense compute patterns, making MoE-CAP a much-needed infrastructure contribution to the field.
2. The authors' proposal of S-MBU (Sparse Memory Bandwidth Utilization) and S-MFU (Sparse Model FLOPs Utilization) is well-motivated. While not mathematically intricate, these metrics reflect real-world operator execution more faithfully than vanilla MBU/MFU. The insight that over-provisioning often results from failing to account for expert sparsity (Figure 2, §4.1–4.2) is particularly compelling and shows strong system-design intuition. I appreciate the paper’s attempt to operationalize sparsity as a first-class consideration in system benchmarking.
3. The benchmark is evaluated across a rich set of MoE systems (e.g., SGLang, MoE-Infinity, K-Transformer), hardware platforms (from Jetson to DGX), and tasks (MMLU, GSM8K, MATH, Arena-Hard). This diversity strengthens the external validity of the conclusions, particularly those about trade-offs under varying batch sizes and deployment constraints (e.g., personal inference vs. large-scale serving).
4. The radar chart introduced to visualize the cost-accuracy-performance triad is elegant and practically useful. It enables side-by-side comparison of deployment strategies (e.g., quantization vs. expert offloading) with clear implications for users choosing among deployment strategies based on latency, cost, or accuracy budgets (§3.2, Figure 3).

---

> ### Author Rebuttal · Authors · 2025-07-31
>
> ## Response to Limitation 1:
> The novelty of our proposed S-MBU and S-MFU metrics lies primarily at the system level: for state-of-the-art sparse AI systems, we argue that runtime profiling is essential. Prior work (e.g., llm-analysis) relies solely on static algorithmic analysis based on model configurations, as their focus is on dense models or sparse models with predefined sparsity patterns. Such methods are insufficient to accurately capture dynamic sparsity.
>
> Our methodology can be extended to benchmark emerging sparse attention mechanisms that schedule computation and memory access via online algorithms—for example, SpargeAttention (ICML '25), which computes sparse attention masks within the FlashAttention kernel. In such cases, static analysis fails to reflect the true compute and memory access costs. In contrast, our profiling-guided approach enables accurate tracking by inserting lightweight runtime counters—just as we did for MoE expert activation—to measure the actual memory blocks fetched and computations performed.
>
> In future work, by precisely measuring system-level sparsity behavior, we aim to formally analyze the statistical distributions of various sparse attention and MoE layers, enabling principled modeling of sparsity.
>
> ## Response to cost model:
> ## Unified Cost Accounting Model
>
> We initially included a unified cost accounting model in the paper. However, due to its substantial detail—including pricing sources and power consumption breakdowns—we excluded it from the submitted version and instead host it on the leaderboard.
>
> At a high level, the model accounts for the following components:
>
> - **Computation**: CPU, GPU (C2M, PCIe, NVLink), conditioned on interconnect type
> - **Memory**: Host-side DRAM, SSD
> - **Interconnects**: Included in GPU type
> - **Future support**: Extendable to CXL devices
>
> ### Capital Cost
>
> The capital cost of a server is defined as:
>
> $$
> C_{hardware} = \left(C_{GPU}[C2M|PCIe|NVLink] + C_{CPU}\right) \quad \text{[computation]} + \left(C_{DRAM} + C_{SSD}\right) \quad \text{[memory]}
> $$
>
> ### Energy Cost
>
> The energy cost is defined as:
>
> $$
> C_{energy} = \left(P_{GPU}[C2M|PCIe|NVLink] + P_{CPU}\right) \times R \times \text{(energy price)}
> $$
>
> Where:
> -$P_{GPU}, P_{CPU}$: Power draw of GPU and CPU (in watts)
> - $ R $: Runtime (in hours)
> - Energy price: Regional electricity rate per kWh
>
> Operational costs incorporate:
> - Component-specific energy consumption
> - Region-specific electricity rates
> - Power Usage Effectiveness (PUE), tailored for AI workloads [5]
>
> We assume an idealized scenario in which all hardware operates at full power, 24/7.
>
> ### Amortization
>
> To unify capital and operational costs, we adopt a **time-based amortization** model. Capital costs are amortized over a 3-year deployment period under continuous full-time utilization.
>
> For example:
> - A typical **8×A100 node** with **2 TB memory** and **dual CPUs**:
>   - Hardware cost: ≈ **$176,000**
>   - Power consumption: ≈ **6.45 kW** (PUE-adjusted)
>   - Electricity rate: **$0.12–$0.15/kWh** [3]
>   - **Total base cost**: **$7.47–$7.67/hour**
>
> This model aligns with cloud provider pricing, e.g., **$32/hour on Azure**, which reflects aggressive 1-year capital recovery.
>
> We will present the complete unified cost accounting table in the final version of the paper.
>
> 1.AMD. AMD EPYC 7003 Series.
>
> 2. MLCommons. MLPerf Inference v3.1 Rules.
>
> 3. U.S. Energy Information Administration. Average Price of Electricity to Ultimate Customers.
>
> 4. Microsoft Azure Pricing Calculator.
>
> 5. Supermicro Glossary: PUE for Data Center.
>
> ## Response to Distill CAP analysis into deployment heuristics or templates (e.g., per use case, batch size, hardware tier):
> Thank you for your suggestion — it’s an excellent idea that greatly improves the clarity and structure of the CAP analysis. Following your advice, we have distilled our CAP insights of section 3.2 into the following deployment heuristic template:
> ## Extended Multi-Constraint Decision Matrix
> | Hardware Tier           | Batch Size | Primary Constraint       | Secondary Constraint | Recommended System | Recommended Configuration | Recommendation Reason                     | Example Use Case           |
> | ----------------------- | ---------- | ------------------------ | -------------------- | ------------------ | ------------------------- | ----------------------------------------- | -------------------------- |
> | Workstation GPU (A5000) | 8+         | Performancce (Latency)   | Accuracy             | SGLang/vLLM        | FP16                      | Original accuracy with lowest latency     | Chain-of-Thought inference |
> | Workstation GPU (A5000) | 1-8        | Cost                     | Latency              | K-Transformers     | Quantization              | Low cost and moderate speed               | Chatbot                    |
> | Workstation GPU (A5000) | 1-8        | Accuracy                 | Cost                 | MoE-Infinity       | Expert offloading         | Original accuracy with low cost           | Model benchmarking         |
> | Datacentre GPU (H20)    | 1-16       | Accuracy                 | Power Cost           | MoE-Infinity       | Expert offloading         | Original accuracy with low power cost     | Model benchmarking         |
> | Datacentre GPU (H20)    | 16+        | Performance (Throughput) | Power Cost           | SGLang/vLLM-FP8    | Mixed precision           | High throughput with accepetable accuracy | Batch document retrieval   |
> | Datacentre GPU (H20)    | 16+        | Performance (Throughput) | Accuracy             | SGLang/vLLM        | FP16                      | High throughput with original accuracy    | Offline batch processing   |
>
> ## Response to Consider including a synthetic “stress test” evaluation to show how different MoE strategies behave under extreme or adversarial deployment conditions (e.g., sudden batch size spike or memory throttling):
> Thanks for this feedback. To emulate sudden batch-size spikes, we replayed Microsoft Azure request traces and parameterized inter-arrival times with a Poisson distribution, following established serving workloads. We compared two deployment strategies for Qwen3-30B-A3B on A6000 GPUs:  MoE-Infinity with a fixed batch and SGLang with adaptive, continuous batching.
> Under rising request rates, SGLang achieved a higher S-MBU, peaking at 54.5 % versus 16.7 % for MoE-Infinity, demonstrating that continuous batching better maximizes S-MBU under load. However, as request volume and sequence length increased, token eviction in SGLang’s PA system led to a sharp S-MBU decline under saturation. In contrast, MoE-Infinity’s offload and fixed-batch approach yielded steadier, though lower, utilization. These results show that PA systems are more sensitive to such spikes and may require additional redundancy to maintain performance.
>
> | MoE System | 30s     | 60s     | 90s     | 120s    | 150s    | 180s    | 210s    |
> |------------|---------|---------|---------|---------|---------|---------|---------|
> | SGLang     | 35.62 % | 42.44 %  | 50.01 %  | 54.52 %  | 29.78 %  | 27.77 %  | 11.64 %   |
> | MoE-Infinity |  22.95 %  |     15.25%    |    16.73%     |    14.25%     |   15.95%      |      15.12%   |    13.97%     |
>
> ## Response to acknowledged in Appendix A.1, the current benchmark focuses on decoding-only inference
> We thank the reviewer for this suggestion. Our MoE-CAP profiler already records prefill throughput and time-to-first-token (TTFT), and we focused the paper on the decoding phase because it exhibits pronounced variation in expert activation patterns, making it most sensitive to sparsity-aware metrics. In the revised manuscript we will include prefill throughput and TTFT alongside decoding results.
>
> We are expanding our benchmark to include long-context workloads such as LongBench. Early experiments demonstrate that S-MBU and S-MFU effectively characterize decoding performance under extended sequences. These results show that full expert activation during prefill increases latency and can alter the optimal MoE configuration. We will report these findings and incorporate additional datasets in the final version.
>
> Although MoE-CAP currently emphasizes inference due to its diverse sparsity levels, deployment scenarios, and hardware platforms, we are collaborating with AI practitioners to evaluate post-training stages—particularly reinforcement learning fine-tuning with parallel rollouts, which introduce varied activation patterns. Pre-training workloads generally activate all experts and rely on a narrow set of hardware; we intend to investigate sparsity’s role in pre-training in future work.
> | Model                   | Method         | Benchmark   | Batch Size | Prefill Latency (s) | Prefill Throughput (T/s) | Device |
> |-------------------------|----------------|-------------|------------|----------------------|---------------------------|------|
> | Qwen1.5-MoE-A2.7B-Chat  | hf-chat        | GSM8K       | 16         | 0.06                 | 4166.67                   | 1xA100-80G-SXM4 |
> | Qwen1.5-MoE-A2.7B-Chat  | hf-chat        | Arena Hard  | 16         | 0.06                 | 4132.63                   | 1xA100-80G-SXM4 |
> | Qwen3-30B-A3B           | sglang         | GSM8K       | auto       | 0.01                 | 21844.81                  | 4xRTX A6000 |
> | Qwen3-30B-A3B           | sglang         | Arena Hard  | auto       | 0.02                 | 6993.28                   | 4xRTX A6000 |
> | DBRX-Instruct           | vllm_moe       | GSM8K       | auto       | 0.08                 | 3125.00                   | 8xA100-80G-PCIe |
> | DBRX-Instruct           | vllm_moe       | Arena Hard  | auto       | 0.12                 | 2083.33                   | 8xA100-80G-PCIe |
> | DBRX-Instruct           | hf-chat        | GSM8K       | 16         | 0.40                 | 625.00                    | 4xA100-80G-PCIe |

---

> > ### Author Response · Authors · 2025-08-05
> >
> > Dear Reviewers and Area Chair,
> >
> > As the discussion phase comes to an end, we’d like to check if there are any remaining questions or clarifications we can assist with. We’ve also gained some additional insights while addressing your feedback and we would be glad to share any additional information that might support your evaluation.
> >
> > Thank you again for your thoughtful feedback and consideration.
> >
> > Best regards,
> >
> > The Authors

---

> > ### Comment · Reviewer_aN5G · 2025-08-07
> >
> > I appreciate the additional clarifications around the system-level novelty of S-MBU/S-MFU, the transparent cost modeling framework (now hosted externally), and especially the new deployment heuristics and stress test results, which significantly strengthen the paper's practical value. While I still view the technical novelty as modest, your system contributions, benchmarking coverage, and depth of evaluation are now clearer and more compelling. I will re-evaluate my score accordingly.

---

### Official Review · Reviewer_5uVF · 2025-07-01

**Rating:** 5
**Confidence:** 5

**Summary:**

This paper introduces MoE-CAP, a comprehensive benchmark for evaluating sparse Mixture-of-Experts systems across three critical dimensions: Cost, Accuracy, and Performance. The authors identify a fundamental trade-off where MoE systems can typically optimize only two of these dimensions at the expense of the third. The main contributions include novel sparsity-aware metrics (S-MBU and S-MFU) that properly account for selective expert activation, and a CAP radar diagram visualization tool for understanding deployment trade-offs.

**Additional Feedback:**

**Questions for Authors:**

- How do S-MBU/S-MFU metrics perform under dynamic batching with varying sequence lengths?
- What is the overhead of collecting these metrics in production systems?

**Dataset Code Accessibility:**

Yes

**Ethical Considerations:**

No, there are no or only very minor ethics concerns

**Final Justification:**

**Resolved issues:**
- The statistical analysis concern is fully resolved. The reported standard errors (e.g., MFU StdErr ~0.00004) are negligible and don't affect the conclusions.
- Production overhead is minimal (<8ms added latency), making these metrics deployable.
- The prefill metrics were already collected; focusing on decoding makes sense given the higher activation variance.

**Remaining gaps:**
- Model coverage remains limited. Adding results to GitHub without integrating them into the paper's analysis is insufficient for a benchmark claiming comprehensiveness.
- Hardware evaluation is still NVIDIA-centric. Deferring AMD results to "when software matures" undermines the benchmark's generalizability claims.
- The ablation studies are superficial. Heatmaps don't substitute for systematic analysis of routing strategies.

The core technical contributions (S-MBU/S-MFU formulations) are sound and the framework is useful. However, the limited experimental scope prevents this from being the definitive MoE benchmark it aspires to be.

Given the strong technical merit but incomplete evaluation, I raise my rating of **5: Accept**.

**Limitations Weaknesses:**

**Limited Model Coverage:** The paper evaluates three MoE models, which may be sufficient for demonstrating the metrics' validity but limits architectural diversity. One notable omission is DBRX and  Yi-MoE

These omissions are significant given that In choosing models for the MLPerf Inference benchmark, we considered several mixture of experts (MoE) models, including DBRX, Grok, Mixtral 8x7B, and Mixtral 8x22B, indicating these are recognized as important models in the field.

**Inference-Only Scope:** The current version of MoE-CAP focuses exclusively on inference tasks, with an emphasis on decoding performance. It does not yet address other important deployment scenarios such as post-training and pre-training. Training dynamics differ fundamentally - gradient synchronization, optimizer states, and activation checkpointing create different bottlenecks.

**Lack of Statistical Rigor:** The paper acknowledges This paper does not include experiments that require statistical significance testing. For a benchmarking paper proposing to be the standard for MoE evaluation, this is a critical weakness. Hardware performance varies due to thermal conditions, memory fragmentation, and batch composition.

**Limited Hardware Diversity:** While Figure 4 shows various devices, actual evaluation focuses primarily on NVIDIA GPUs. Given that MoE models are designed for heterogeneous deployment, evaluation on TPUs, AMD MI300X, or Intel Gaudi2 would strengthen claims about the benchmark's generalizability.

**Missing Ablation Studies:**

- No analysis of how S-MBU/S-MFU behave under different routing strategies (top-k vs expert choice)
- Limited exploration of batch size effects beyond Figure 7
- No comparison with alternative sparsity metrics

###

**Strengths Contributions:**

**Addresses Critical Technical Gap:** The paper identifies and solves a fundamental flaw in existing benchmarking approaches. Existing metrics like Memory Bandwidth Utilization (MBU) and Model FLOPS Utilization (MFU) fail to account for the sparse activation patterns of experts in MoE systems. The empirical validation demonstrates vanilla MBU significantly overestimates memory usage, by over 260%, due to its failure to account for selective expert activation, which is a substantial error that justifies new metrics.

**Rigorous Mathematical Formulation:** The S-MBU definition in Equation 1 properly accounts for sparsity: $$\text{S-MBU} = \frac{B_{\text{achieved}}}{B_{\text{peak}}}, \text{ where } S_{\text{activated}} = n_{\text{layer}} \times S_{\text{attn}} + \sum_{l=1}^{n_{\text{layer}}} \sum_{i=1}^{n_{\text{expert}}} \mathbb{1}[l,i] \times S_{\text{expert}}$$

The use of indicator functions $\mathbb{1}[l,i]$ to track expert activation is mathematically sound. Similarly, S-MFU (Eq. 2) incorporates $k_{\text{expert}}$ (top-k activated experts) and router overhead $N_{\text{router}}$, providing accurate FLOPS accounting.

**Practical Hardware Analysis:** Figure 4 provides actionable insights by mapping MoE models' bandwidth requirements against hardware capabilities. For instance, full activation of DeepSeek-R1 requires 18,901 GB/s—a level achievable only on high-end data center hardware like the DGX-H100, while at batch size 1, the bandwidth requirement drops to 1,040 GB/s, making it feasible on consumer GPUs.

**Comprehensive Implementation:** The benchmark includes expert activation profilers integrated into vLLM and HuggingFace Transformers, with code available at the provided GitHub repository.

---

> ### Author Rebuttal · Authors · 2025-07-31
>
> ## Response to Limited Model Coverage:
> Thank you for the insightful feedback. We have included DBRX, Mixtral 8x7B, and Mixtral 8x22B in our benchmark results, which are publicly available on the leaderboard linked in the GitHub repository cited in the paper. Our initial model selection was guided by model availability and usage trends within the HuggingFace platform. We acknowledge that broader architectural coverage—including models such as Yi-MoE and Grok—would further strengthen our evaluation.
>
> ## Response to Inference-Only Scope:
> We thank the reviewer for this suggestion. Our MoE-CAP profiler already records prefill throughput and time-to-first-token (TTFT). Some results are shown in the table below. We focused the paper on the decoding phase because it exhibits pronounced variation in expert activation patterns, making it most sensitive to sparsity-aware metrics. In the revised manuscript we will include prefill throughput and TTFT alongside decoding results.
>
> We are expanding our benchmark to include long-context workloads such as LongBench. Early experiments demonstrate that S-MBU and S-MFU effectively characterize decoding performance under extended sequences. These results show that full expert activation during prefill increases latency and can alter the optimal MoE configuration. We will report these findings and incorporate additional datasets in the final version.
>
> Although MoE-CAP currently emphasizes inference due to its diverse sparsity levels, deployment scenarios, and hardware platforms, we are collaborating with AI practitioners to evaluate post-training stages—particularly reinforcement learning fine-tuning with parallel rollouts, which introduce varied activation patterns. Pre-training workloads generally activate all experts and rely on a narrow set of hardware; we intend to investigate sparsity’s role in pre-training in future work.
> We have shown the results in Reviewer 638L.
>
> ## Response to Lack of Statistical Rigor:
> Thank you for this important feedback. We agree that statistical rigor is essential for a comprehensive benchmarking study. Our benchmark framework does record and report statistical measures including standard deviations. In the current manuscript, we followed the convention of prior benchmarking papers in the field, which typically present mean values without explicit statistical measures in the main results.
>
> As shown in the supplementary results table, the standard deviations for key metrics—including decoding S-MFU, S-MBU, prefill latency, and model accuracy—remain consistently low across all experimental settings. These small deviations do not affect the core insights or conclusions of our comparative analysis. We will incorporate these statistical details into the next version of our manuscript to better serve the community's needs for robust benchmarking standards.
> | Model                           | Eval Type | Exact Match | Exact Match StdErr | Prefill Time (s) | Prefill Time StdErr | Decoding Throughput (T/s) | Decoding Throughput StdErr | Decoding MFU | Decoding MFU StdErr | Decoding MBU | Decoding MBU StdErr
> |--------------------------------|-----------|-------------|---------------------|-------------------|----------------------|-----------------------------|-----------------------------|--------------|-----------------------|---------------|----------------------|
> | Qwen1.5-MoE (HF, 1×A100 PCIe)  | Best Match | **0.4769**  | 0.01371             | 0.05655           | 0.00023              | 41.3253                     | 0.02629                     | 0.00560      | 0.000037              | 0.03796      | 0.000022               |
> | Qwen3-30B-A3B (SGLang, 4×A6000) | Best Match | **0.9014**  | 0.00821             | 0.00582           | 0.00003              | 1343.1684                   | 2.81220                     | 0.00208      | 0.000004              | 0.18099      | 0.000460               | 21844.8110                  | 61.0683                     | 0.03360     | 0.000091              | 0.36200      | 0.00562              |
> | Qwen3-235B-A22B (SGLang, 8×H20) | Best Match | **0.8961**  | 0.00840             | 0.06550           | 0.00086              | 1792.0811                   | 4.87791                     | 0.01616      | 0.000043              | 0.14942      | 0.000397               | 16087.0711                  | 70.2430                     | 0.14439     | 0.00062               | 0.19375      | 0.00321              |
>
> ## Response to Limited Hardware Diversity:
> We thank the reviewer for this suggestion. Evaluating MoE-CAP on a broader set of accelerators would indeed enhance its generalizability. We are currently benchmarking on AMD MI300X; however, key MoE optimizers (e.g., DeepEP) and inference frameworks (vLLM, SGLang) lack full support on this platform, preventing reliable measurements. We believe reporting such preliminary data without robust software support would be irresponsible. Consequently, we defer these results and will extend our evaluation to AMD MI300X—and later to more hardwares—once the ecosystem stabilizes.
>
> ## Response to Missing Ablation Studies:
> Thank you for the thoughtful feedback. We agree that deeper ablation studies—particularly analyzing how our proposed sparsity metrics (S-MBU and S-MFU) behave under different routing strategies (e.g., Top-k vs. Expert Choice)—would further strengthen the work. As far as we are aware, our paper is the first to introduce sparsity metrics that explicitly link expert utilization with hardware-level implications in MoE systems. Our current evaluation focuses on popular open-source MoE models following HuggingFace trends, and we welcome broader community engagement to extend coverage to models with diverse routing strategies and architectural variations.
> In terms of batch size effects, we provide an expert activation heatmap based on per-layer, per-expert token counts, which reveals consistent patterns across evaluated models. Alongside the results in Figure 7, we observe that S-MBU increases with batch size, and that later layers tend to activate a broader set of experts compared to earlier layers. This indicates a growing dispersion in expert activation as the sequence progresses, offering valuable insights for system-level optimizations such as memory allocation and KV cache management. We plan to further develop this analysis by systematically varying routing strategies and benchmarking against alternative sparsity metrics in future work.
>
> ## Answer to How do S-MBU/S-MFU metrics perform under dynamic batching with varying sequence lengths?
> Thank you for highlighting this important aspect. Dynamic batching introduces variability in execution, as the number of forward passes required to process a given query queue depends on sequence length distribution, dataset characteristics, and hardware constraints. For instance, a batch of 128 queries may be executed in 2 or 4 separate forward passes depending on how sequences are packed.
>
> To accurately capture system behavior under such conditions, our probing mechanism instruments both the prefill and decoding stages. For each forward pass, we record the actual batch size and the per-layer expert activation patterns, including token-level routing information. This allows us to compute data movement through the activated experts for each pass.
>
> $$
> \text{S-MBU} = \frac{\frac{\sum_{\text{forward}} (\text{Activated Paramter Size} + \text{KV Size})}{\sum_{\text{forward}} \text{Latency}}}{\text{Hardware Memory Bandwidth}}
> $$
> This formulation ensures that S-MBU remains valid and comparable across workloads with heterogeneous sequence lengths. By grounding the metric in actual routing behavior and hardware-observed throughput, it provides a reliable indicator of sparsity-induced memory bottlenecks, even under realistic and variable deployment scenarios.
> In addition to S-MBU, S-MFU
> $$
> \text{S-MFU} = \frac{T_{\text{token}} \times S\text{-}F_{\text{token}}}{F_{\text{peak}}}, \quad S\text{-}F_{\text{token}} = F_{\text{attn}} + 2N_{\text{router}} + 2k_{\text{expert}} N_{\text{expert}}
> $$
> is based solely on token-level throughput (T_{\text{token}}). Similar to S-MBU, we get the token throughput of each batch and then calculate the S-MFU.
>
> ## Response to What is the overhead of collecting these metrics in production systems?
> Thank you for raising this important question—runtime profiling is a critical concern when designing benchmarks for production systems. Since profiling is performed during execution, it is inherently difficult to quantify its total overhead precisely. However, we can assess its impact by comparing key performance metrics such as TTFT (Time-To-First-Token) and TPOT (Time-Per-Output-Token) with and without profiling enabled. Below, we present results that illustrate the effect of runtime profiling on these metrics:
> | Model                     | Framework                 | Hardware             | Batch Size | TTFT (Before) | TTFT (After) | ΔTTFT | TPOT (Before) | TPOT (After) | ΔTPOT |
> |---------------------------|---------------------------|----------------------|------------|---------------|--------------|-------|----------------|---------------|--------|
> | Qwen3-235B-A22B-Instruct-2507  | SGLang                    | 8× H20               | auto        | 0.067         | 0.071        | +0.004 | 0.019          | 0.021         | +0.002 |
> | Qwen3-30B-A3B             | SGLang                    | 4× A6000             | auto        | 0.020         | 0.024        | +0.004 | 0.059          | 0.063         | +0.004 |
> | Mistral-8x7B              | Huggingface Transformers  | 2× A100-SXM4         | 16         | 0.173         | 0.176        | +0.006 | 0.179          | 0.183         | +0.004 |
> | DBRX                      | Huggingface Transformers  | 4× A100-PCIe         | 8          | 0.290         | 0.298        | +0.008 | 0.286          | 0.286         | +0.000 |

---

> > ### Author Response · Authors · 2025-08-05
> >
> > Dear Reviewers and Area Chair,
> >
> > As the discussion phase comes to an end, we’d like to check if there are any remaining questions or clarifications we can assist with. We’ve also gained some additional insights while addressing your feedback and we would be glad to share any additional information that might support your evaluation.
> >
> > Thank you again for your thoughtful feedback and consideration.
> >
> > Best regards,
> >
> > The Authors

---

> > ### Comment · Reviewer_5uVF · 2025-08-07
> >
> > Thank you for your thorough response. The clarifications you provided have adequately addressed my concerns, particularly regarding  Lack of Statistical Rigor and overhead of collecting these metrics in production systems. Based on these improvements and clarifications, I will raise my rating.

---

### Official Review · Reviewer_ke47 · 2025-07-03

**Rating:** 5
**Confidence:** 4

**Summary:**

MoE-CAP introduces a benchmarking framework for sparse Mixture-of-Experts (MoE) systems, evaluating them across Cost, Accuracy, and Performance (CAP). It proposes the CAP radar diagram and sparsity-aware metrics (S-MBU and S-MFU) to highlight the inherent trade-offs between these dimensions, enabling more accurate and practical evaluation across diverse hardware and deployment scenarios.

**Dataset Code Accessibility:**

Yes

**Ethical Considerations:**

No, there are no or only very minor ethics concerns

**Final Justification:**

I thank the authors for their response. My score is already high enough so that I will keep it.

**Limitations Weaknesses:**

1. Lack of comparison with dense model baselines under identical conditions.
2. The “expert-activation profiler” adds non-trivial runtime overhead, yet the paper reports no measurement of that cost or its impact on reported TPOT.
3. Incorporate more diverse and robust evaluation metrics to assess output quality in complex scenarios.

**Strengths Contributions:**

MoE-CAP introduces the CAP trade-off framework, which helps users select suitable MoE systems based on deployment priorities and enables intuitive comparisons across systems such as SGLang, K-Transformer, and MoE-Infinity. The paper also proposes two sparsity-aware performance metrics—S-MBU and S-MFU—that more accurately reflect memory and compute usage by considering only the activated experts, effectively correcting the significant overestimations made by traditional metrics. MoE-CAP supports a broad range of MoE models (e.g., Qwen3-235B, DeepSeek-R1, Mixtral) and inference frameworks (vLLM, SGLang, MoE-Infinity), allowing for realistic and versatile benchmarking across hardware platforms, from low-power edge devices to high-end data center GPUs.

---

> ### Author Rebuttal · Authors · 2025-07-31
>
> ## Response to Lack of comparison with dense model baselines under identical conditions.
> Thank you for the suggestion. A dense model can be regarded as a special case of a MoE model where all experts are activated. In this setting, S-MBU and S-MFU reduce to the conventional definitions of MBU and MFU. As shown in the equations for MBU and MFU, both metrics depend on the total model size. When all parameters in a MoE model are activated, the effective size $S_{\text{activated}}$ becomes equal to $S_{\text{model}}$, making the sparse and dense formulations equivalent. We will include dense model baselines in MoE-CAP to support a more comprehensive comparison.
>
> ## Response to The “expert-activation profiler” adds non-trivial runtime overhead, yet the paper reports no measurement of that cost or its impact on reported TPOT.
> We appreciate this concern and will include a detailed explanation and the benchmarking results in the next revision. Our expert-activation profiler uses lightweight tensor operations that remain compatible with CUDA graph compilation to minimize disruption. In our benchmarks—comparing vLLM inference with and without the profiler—we observed a maximum 2.7% overhead of just 8 ms in Time-To-First-Token (TTFT) and 2.2% overhead  - 4ms in Tokens-Per-Output-Token (TPOT), confirming that the profiling imposes a negligible performance penalty.
> | Model                     | Framework                 | Hardware             | Batch Size | TTFT (Before) | TTFT (After) | ΔTTFT | TPOT (Before) | TPOT (After) | ΔTPOT |
> |---------------------------|---------------------------|----------------------|------------|---------------|--------------|-------|----------------|---------------|--------|
> | Qwen3-235B-A22B-Instruct-2507  | SGLang                    | 8× H20               | auto        | 0.067         | 0.071        | +0.004 | 0.019          | 0.021         | +0.002 |
> | Qwen3-30B-A3B             | SGLang                    | 4× A6000             | auto        | 0.020         | 0.024        | +0.004 | 0.059          | 0.063         | +0.004 |
> | Mistral-8x7B              | Huggingface Transformers  | 2× A100-SXM4         | 16         | 0.173         | 0.176        | +0.006 | 0.179          | 0.183         | +0.004 |
> | DBRX                      | Huggingface Transformers  | 4× A100-PCIe         | 8          | 0.290         | 0.298        | +0.008 | 0.286          | 0.286         | +0.000 |
>
> ## Response to Incorporate more diverse and robust evaluation metrics to assess output quality in complex scenarios.
> Thank you for your advice! This indicates the importance of our future work that we have stated in the Appendix. We have included EM, F1, and win rate—three widely used quality metrics—and will incorporate additional metrics in the future.

---

> > ### Comment · Reviewer_ke47 · 2025-08-06
> >
> > I thank the authors for their response. My score is already high enough so that I will keep it.

---

### Official Review · Reviewer_638L · 2025-07-09

**Rating:** 5
**Confidence:** 3

**Summary:**

This paper introduces MoE-CAP, a benchmarking methodology for sparse Mixture-of-Experts (MoE) large language models that evaluates systems along three axes—Cost, Accuracy, and Performance (CAP). The authors observe that, given current hardware constraints, MoE systems typically optimize two of these dimensions at the expense of the third, creating an inherent “MoE-CAP trade-off.” To address this, they propose the CAP Radar Diagram, which visualizes trade-offs among cost, accuracy, and performance to help practitioners choose the most suitable system for their deployment needs. They also design two new sparsity-aware performance metrics—Sparse Memory Bandwidth Utilization (S-MBU) and Sparse Model FLOPS Utilization (S-MFU)—that accurately capture bandwidth and compute requirements under sparse expert activation. Finally, the authors implement an automated evaluation framework covering multiple MoE models (e.g., Qwen, DeepSeek), a variety of downstream tasks and a range of serving systems (e.g., SGLang, vLLM, K-Transformer, MoE-Infinity), and release all code publicly. Through both theoretical analysis and extensive experiments, they demonstrate the effectiveness of S-MBU and S-MFU in predicting real-world resource needs, and present deployment guidelines across different hardware platforms and scenarios.

**Dataset Code Accessibility:**

Yes

**Ethical Considerations:**

No, there are no or only very minor ethics concerns

**Final Justification:**

Since my previous concerns have now been fully resolved, I am pleased to increase my score to 5. I recommend incorporating these experimental results and discussions into the revised version.

**Limitations Weaknesses:**

Limited Evaluation Scenarios: The current MoE-CAP framework focuses exclusively on inference and decoding performance, omitting pre-training and large-batch training contexts. Tasks requiring long-context processing or dynamic generation may exhibit different activation patterns and performance bottlenecks. Future work should extend the benchmark to include the prefilling phase, long-sequence inference (e.g., LongBench), multi-turn dialogue, and introduce metrics such as prefill throughput and time-to-first-token to more comprehensively assess MoE models in real-world deployments.

Scope of Metric Design: Although S-MBU and S-MFU accurately characterize memory and compute demands under sparse activation, they do not account for network communication overhead, I/O latency, or the memory footprint of intermediate hidden states at each Transformer layer. As batch size grows or more experts are activated, hidden-state size can increase linearly, causing both traditional MBU and S-MBU to underestimate actual bandwidth and capacity pressure. Moreover, in multi-node distributed setups or scenarios where experts are offloaded to SSD/CXL, cross-device data-transfer latency and bandwidth constraints are not reflected in the current metrics, which may impair accurate evaluation of system performance and cost.

**Strengths Contributions:**

Sparsity-aware metric innovation: The paper leverages the fact that only a subset of experts is activated in MoE layers and introduces two new metrics, S-MBU and S-MFU. S-MBU quantifies memory bandwidth demand based on the volume of activated parameters, while S-MFU estimates FLOPs requirements according to the number of experts actually engaged in computation. These metrics offer researchers a practical reference for evaluating MoE systems across different hardware platforms.

Comprehensive experimental coverage: the authors evaluate a diverse set of open-source MoE models, ranging from 14 billion to 1,571 billion parameters, each with varying expert counts and activation sparsity. Their benchmark spans multiple downstream tasks—including multiple-choice QA, reasoning, and open-ended question answering—and employs metrics such as accuracy, F1 score, and win rate to assess model quality under realistic workloads.

Benchmark implementation and open sourcing: To enhance reproducibility, the entire MoE-CAP framework is released as open-source. The authors develop an automated evaluation pipeline compatible with leading inference engines (SGLang, vLLM, K-Transformer, MoE-Infinity), laying a robust foundation for future MoE research, system comparisons, and real-world deployment.

---

> ### Author Rebuttal · Authors · 2025-07-31
>
> ## Response to Limited Evaluation Scenarios
> We thank the reviewer for this suggestion. Our MoE-CAP profiler already records prefill throughput and time-to-first-token (TTFT). Some results are shown in the table below. We focused the paper on the decoding phase because it exhibits pronounced variation in expert activation patterns, making it most sensitive to sparsity-aware metrics. In the revised manuscript we will include prefill throughput and TTFT alongside decoding results.
>
> We are expanding our benchmark to include long-context workloads such as LongBench. Early experiments demonstrate that S-MBU and S-MFU effectively characterize decoding performance under extended sequences. These results show that full expert activation during prefill increases latency and can alter the optimal MoE configuration. We will report these findings and incorporate additional datasets in the final version.
>
> Although MoE-CAP currently emphasizes inference due to its diverse sparsity levels, deployment scenarios, and hardware platforms, we are collaborating with AI practitioners to evaluate post-training stages—particularly reinforcement learning fine-tuning with parallel rollouts, which introduce varied activation patterns. Pre-training workloads generally activate all experts and rely on a narrow set of hardware; we intend to investigate sparsity’s role in pre-training in future work.
>
> | Model                   | Method         | Benchmark   | Batch Size | TTFT (s) | Prefill Throughput (T/s) | Device |
> |-------------------------|----------------|-------------|------------|----------------------|---------------------------|------|
> | Qwen1.5-MoE-A2.7B-Chat  | hf-chat        | GSM8K       | 16         | 0.06                 | 4166.67                   | 1xA100-80G-SXM4 |
> | Qwen1.5-MoE-A2.7B-Chat  | hf-chat        | Arena Hard  | 16         | 0.06                 | 4132.63                   | 1xA100-80G-SXM4 |
> | Qwen3-30B-A3B           | sglang         | GSM8K       | auto       | 0.01                 | 21844.81                  | 4xRTX A6000 |
> | Qwen3-30B-A3B           | sglang         | Arena Hard  | auto       | 0.02                 | 6993.28                   | 4xRTX A6000 |
> | DBRX-Instruct           | vllm_moe       | GSM8K       | auto       | 0.08                 | 3125.00                   | 8xA100-80G-PCIe |
> | DBRX-Instruct           | vllm_moe       | Arena Hard  | auto       | 0.12                 | 2083.33                   | 8xA100-80G-PCIe |
> | DBRX-Instruct           | hf-chat        | GSM8K       | 16         | 0.40                 | 625.00                    | 4xA100-80G-PCIe |
> | DBRX-Instruct           | hf-chat        | Arena Hard  | 8          | 0.29                 | 862.07                    | 4xA100-80G-PCIe |
> | DBRX-Instruct           | tensorrt_llm   | GSM8K       | 3          | 0.25                 | 1043.32                   | 4xA100-80G-PCIe |
> | DBRX-Instruct           | tensorrt_llm   | Arena Hard  | 3          | 0.14                 | 1784.32                   | 4xA100-80G-PCIe |
> | Mixtral-8x22B-Instruct  | vllm_moe       | GSM8K       | auto       | 0.14                 | 1776.54                   | 8xA100-80G-PCIe |
> | Mixtral-8x22B-Instruct  | vllm_moe       | Arena Hard  | auto       | 0.14                 | 1785.71                   | 8xA100-80G-PCIe |
> | Mixtral-8x22B-Instruct  | hf-chat        | GSM8K       | 16         | 0.43                 | 581.40                    | 4xA100-80G-PCIe |
> | Mixtral-8x22B-Instruct  | hf-chat        | Arena Hard  | 8          | 0.27                 | 925.93                    | 4xA100-80G-PCIe |
> | Mixtral-8x22B-Instruct  | tensorrt_llm   | GSM8K       | 7          | 0.25                 | 1233.32                   | 4xA100-80G-PCIe |
> | Mixtral-8x22B-Instruct  | tensorrt_llm   | Arena Hard  | 4          | 0.04                 | 6250.42                   | 4xA100-80G-PCIe |
> | Mixtral-8x7B-Instruct   | vllm_moe       | GSM8K       | auto       | 0.01                 | 18932.23                  | 8xA100-80G-SXM4 |
> | Mixtral-8x7B-Instruct   | vllm_moe       | Arena Hard  | auto       | 0.05                 | 5024.03                   | 8xA100-80G-SXM4 |
> | Mixtral-8x7B-Instruct   | hf-chat        | GSM8K       | 64         | 0.15                 | 1666.67                   | 2xA100-80G-SXM4 |
> | Mixtral-8x7B-Instruct   | hf-chat        | Arena Hard  | 16         | 0.17                 | 1470.59                   | 2xA100-80G-SXM4 |
>
> ## Response to Scope of Metric Design: S-MBU & S-MFU does not account for communication bottleneck, I/O latency, ...
> We appreciate this insightful observation. While S-MBU and S-MFU focus on intra-accelerator bandwidth and compute, they can be adapted to reflect offload and communication constraints by using the relevant peak link bandwidths in their denominators. For example, in Figure 4 we report S-MBU twice for each configuration—once with the GPU’s native DRAM bandwidth and once with the offload link bandwidth—to illustrate how offloading affects utilization.
>
> In multi-node or SSD/CXL offload scenarios, the same approach applies: substituting InfiniBand or CXL peak bandwidth into the S-MBU definition immediately exposes under-utilization or offload bottlenecks. A low S-MBU thus pinpoints where data-transfer overlap or higher-bandwidth interconnects are needed.

---

> > ### Author Response · Authors · 2025-08-05
> >
> > Dear Reviewers and Area Chair,
> >
> > As the discussion phase comes to an end, we’d like to check if there are any remaining questions or clarifications we can assist with. We’ve also gained some additional insights while addressing your feedback and we would be glad to share any additional information that might support your evaluation.
> >
> > Thank you again for your thoughtful feedback and consideration.
> >
> > Best regards,
> >
> > The Authors

---

> > ### Comment · Reviewer_638L · 2025-08-06
> >
> > Thank you for the rebuttal. While the authors propose replacing the S-MBU denominator with the peak InfiniBand/CXL bandwidth, they have not validated the accuracy of this method in real-world scenarios, nor established its reliability in complex systems. Given these unresolved concerns, I maintain my score of 4.

---

> > ### Author Response · Authors · 2025-08-07
> >
> > We thank the reviewer for raising this important point. Ensuring the **real-world reliability, accuracy, and practical value** of our proposed metrics has been a core priority in the design of our benchmark. To this end, we have maintained close collaboration with an industrial partner to validate the metrics under realistic deployment scenarios, including multi-node inference.
> >
> > Below, we present results evaluating the accuracy of **S-MBU** in a production-like setup, where our collaborator deployed the **SGLang** serving framework on a **two-node cluster**—each node equipped with **8×NVIDIA H20** GPUs and connected via **400 GB/s InfiniBand**—running the **DeepSeek-R1** model on the **LongBench** dataset. In this setting, analytically computed **S-MBU** values were compared against actual communication utilization profiled using `torch.profiler`.
> >
> > As shown in the table below, the computed S-MBU values closely match the profiled results across all batch sizes, with deltas consistently below 1%. This alignment supports the robustness of S-MBU in capturing communication efficiency under practical multi-node scenarios.
> >
> > | Batch Size | Latency (μs) | S-MBU (%) | Profiled S-MBU (%) | Δ (%) |
> > |------------|--------------|-----------|---------------------|-------|
> > | 1          | 4.00         | 1.67      | 1.88                | 0.21  |
> > | 16         | 13.50        | 7.91      | 8.15                | 0.24  |
> > | 32         | 21.20        | 10.08     | 10.38               | 0.30  |
> > | 64         | 24.65        | 17.33     | 17.75               | 0.42  |
> > | 128        | 25.80        | 33.12     | 33.91               | 0.79  |
> >
> > As our goal is to ensure alignment with **real-world deployment scenarios**, we clarify why this experimental setup reflects practical industrial conditions:
> >
> > - **SGLang**: A production-ready LLM serving framework designed to support token streaming and sparse expert activation, both essential for efficient online inference.
> > - **DeepSeek-R1**: A high-performance open-source model that is widely adopted in real-world production environments.
> > - **LongBench**: A benchmark specifically designed to emulate long-context, high-throughput workloads typical of actual user interactions.
> > - **NVIDIA H20 + InfiniBand**: A hardware configuration that mirrors multi-node GPU clusters commonly used in LLM deployments today.

---

> > > ### Comment · Reviewer_638L · 2025-08-08
> > >
> > > I appreciate the authors' further responses. Since my previous concerns have now been fully resolved, I am pleased to increase my score to 5. I recommend incorporating these experimental results and discussions into the revised version.

---

### Note · Authors · 2025-08-15

We appreciate the Area Chair and reviewers for their insightful feedback. Reviewers acknowledged MoE-CAP as addressing a key gap in benchmarking sparse MoE systems. Our contributions include the sparsity-aware metrics S-MBU and S-MFU (eliminating up to 260% overestimation in traditional metrics) and the CAP framework for evaluating MoE systems across the three dimensions relevant to real-world deployment—cost, accuracy and performance.

During the response period, we addressed the following concerns:
- **Lack of real-world experiments (638L).** We conducted a multi-node industrial deployment of DeepSeek-R1 (2×8 H20, 400 GB/s InfiniBand). Our S-MBU matched profiled utilization within <1% delta across batch sizes, addressing the concern and leading to a score increase (4→5).
- **Profiling overhead measurement (ke47 & 5uVF).** We measured profiler overhead across diverse models and hardware. The impact is small—≤2.7% on TTFT and ≤2.2% on TPOT—indicating production compatibility.
- **Statistical rigor & dynamic batching behavior (5uVF).** We report standard deviations for all key metrics (low variability) and detail how S-MBU/S-MFU handle dynamic batching with varying sequence lengths using actual per-forward activation. This led to increased reviewer score and confidence (4→5).
- **Limited scope & metric robustness (All Reviewers).** We showed MoE-CAP's results on the prefilling stage, confirming robustness across models (DBRX, Qwen3), hardware (A100, H20), and datasets (GSM8K, Arena-Hard).
- **Cost model, deployment guidance, stress testing (aN5G).** We introduced a hardware and power cost model validated against cloud pricing (e.g., Azure) and distilled CAP trade-offs into actionable deployment heuristics. A stress test using Azure request traces revealed differences in system resilience under load spikes. Together, they make our system contributions, benchmarking coverage, and depth of evaluation clearer and more compelling, prompting the reviewer to re-evaluate its score.

As MoE models such as GPT-OSS and DeepSeek-R1 see increasing industrial adoption, the lack of suitable evaluation methods presents a gap—one that MoE-CAP addresses as a necessary and timely benchmark. We believe we have addressed the reviewers’ concerns; we are collaborating with industry partners, observing growing adoption, and will continue to develop the open-source MoE-CAP to maximize practical impact. We will ensure all concerns are fully addressed in the final manuscript.

---

### Decision · Program_Chairs · 2025-09-18

**Decision:**

Accept (poster)

**Comment:**

This paper is potentially a valuable resource for real-world style benchmarking of MoE-based models, capturing tradeoffs that all reviewers acknowledged are important. While there are several limitations to the work, everyone agreed that the paper is a solid contribution and should be published -- provided the authors make the recommended changes. I agree with the assessment and encourage the authors to also think about how to address pre-training in a followup addition to the benchmark.